# Soil moisture monitoring with cosmogenic neutrons: an asset for the development and assessment of soil moisture products in the state of Brandenburg (Germany)

Maik Heistermann[1], Daniel Altdorff[1,2], Till Francke[1], Martin Schrön[2], Peter M. Grosse[1,2], Arvid Markert[3], Albrecht Bauriegel[3], Peter Biró[1], Sabine Attinger[2], Frank Beyrich[4], Peter Dietrich[2], Rebekka Eichstädt[5], Jakob Terschlüsen[1], Ariane Walz[5], Steffen Zacharias[2], and Sascha E. Oswald[1]

[1]Institute of Environmental Science and Geography, University of Potsdam, Potsdam, Germany
[2]UFZ - Helmholtz Centre for Environmental Research GmbH, Permoserstr. 15, Leipzig, Germany
[3]Landesamt für Bergbau, Geologie und Rohstoffe Brandenburg, Inselstraße 26, Cottbus, Germany
[4]Deutscher Wetterdienst, Meteorologisches Observatorium Lindenberg - Richard-Aßmann-Observatorium, Am Observatorium 12, 15848 Tauche, Germany
[5]Ministerium für Wirtschaft, Arbeit, Energie und Klimaschutz (MWAEK), Henning-von-Tresckow-Straße 2-13, Potsdam, Germany

**Correspondence:** Maik Heistermann (maik.heistermann@uni-potsdam.de)

**Abstract.**

In the recent years, the German federal state of Brandenburg has been particularly impacted by soil moisture droughts. To support the timely and informed management of such water-related risks, we introduce a novel soil moisture monitoring network based on cosmic-ray neutron sensing (CRNS) technology. Driven by a joint collaboration of research institutions and federal state agencies, eight sites across Brandenburg were instrumented in 2024; four more are to be deployed in November 2025. The data is openly accessible in order to foster applications and collaboration right from the start. In this paper, we present the network design, evaluate and discuss the CRNS-based soil moisture estimates from 2024 until 2025, and demonstrate how the observations from this network can serve to evaluate and improve soil moisture products with regard to their applicability in Brandenburg. Specifically, we compare selected large-scale products from modelling and remote sensing (ERA5-Land, the Soil Water Index of the Copernicus Land Monitoring Service, and the Copernicus Climate Change Service surface soil moisture product) to the Soil-Water-Atmosphere-Plant (SWAP) model that was set up based on region-specific data. We conclude that model-based products (ERA5-Land and, in particular, the region-specific SWAP model) have the highest potential to mitigate the inherent limitations of a sparse instrumental soil moisture network (such as limited temporal, horizontal and vertical coverage). We further discuss resulting implications for the management of water-related risks in Brandenburg, practical lessons learned from the establishment and operation of the network, as well as potential future applications.

## 1 Introduction

Soil moisture acts as a key state variable in the earth system: it exerts a major control on evapotranspiration, and hence the exchange of water and energy between soil and atmosphere. Furthermore, soil moisture affects the vitality and productivity

of natural vegetation as well as agricultural systems, and influences groundwater recharge, runoff formation, the emission and sequestration of soil organic carbon, as well as wildfire hazards.

The importance of soil moisture, and hence its monitoring, becomes specifically obvious in Brandenburg as one of the driest federal states in Germany. Large parts of the state are governed by relatively low annual precipitation sums (between 500 and 700 mm/a) and permeable sandy soils with low water retention capacity. This combination entails various drought-related hazards which became particularly obvious in the years 2018 to 2022. In this period, Brandenburg was affected by declining groundwater tables (Pohle et al., 2024; Warter et al., 2024), wild fires (Priesner et al., 2024), forest degradation (Horn et al., 2025; Priesner et al., 2024), and crop yield losses (Brill et al., 2024). While locations with a deep groundwater table are particularly prone to drought effects on vegetation, Brandenburg additionally features extensive lowland and wetland areas with shallow groundwater tables. In these areas, evapotranspiration in summer is particularly high, which causes a substantial pressure on water availability in lakes and rivers (Francke and Heistermann, 2025; Pohle et al., 2024; Warter et al., 2024). Again, this process is regulated by root-zone soil moisture.

The usefulness of soil moisture monitoring is widely acknowledged, e.g., for irrigation management (Datta and Taghvaeian, 2023), drought early warning (Satapathy et al., 2024), earth systems modelling (Dorigo et al., 2017; Miralles et al., 2019), climate change impact assessment (IPCC, 2022), or the detection of flash droughts (Li et al., 2023). Still, it remains a notorious challenge to obtain timely and reliable observations at useful spatio-temporal coverage and resolution. Conventional point-scale sensors are invasive and suffer from a lack of spatial representativeness (Blöschl and Grayson, 2000), while remote sensing products are limited by shallow penetration depths, low overpass frequencies, and vegetation-related uncertainties (Babaeian et al., 2019; Li et al., 2021; Peng et al., 2021; Schmidt et al., 2024).

Within the past decade, cosmic-ray neutron sensing (CRNS) has emerged as a promising alternative (Zreda et al., 2012; Andreasen et al., 2017). It allows for continuous and non-invasive monitoring of soil moisture with a measurement depth of tens of centimeters ("the root zone") and a footprint radius of approximately 130–240 m (depending on air humidity, soil moisture, and vegetation, see Köhli et al., 2015). That way, CRNS enables robust estimates that are representative at the scale of agricultural fields, hydrotopes, or typical landscape parcels. In a densely instrumented agricultural research site near Potsdam (Brandenburg), Heistermann et al. (2023) already demonstrated the capability of multiple CRNS sensors to consistently capture the prolonged soil moisture droughts during the years 2019, 2020, and 2022. Due to their large horizontal footprint, CRNS-based soil moisture estimates are being increasingly used to evaluate large-scale soil moisture products from modelling and remote sensing (Vinodkumar et al., 2017; Cooper et al., 2024; Schmidt et al., 2024; Zheng et al., 2024), or for the assimilation into land surface models (Patil et al., 2021; Li et al., 2024; Fatima et al., 2024; Szczykulska et al., 2024). When operated in mobile mode, the spatial extent of the CRNS measurement can be increased towards regional scales, depending on the carrying vehicle (e.g., Franz et al., 2015; McJannet et al., 2017; Schrön et al., 2021; Handwerker et al., 2025). In combination with stationary CRNS, such observations could be compared to state variables of hydrological models in space in time, while, in turn, the nodes of stationary CRNS networks could serve as reference points within the mobile CRNS routes.

So far, long-term national scale soil moisture monitoring networks on the basis of CRNS technology have been deployed only in few countries, e.g., the USA (COSMOS, Zreda et al., 2012), UK (COSMOS-UK, Cooper et al., 2021), and Australia

(CosmOz, Hawdon et al., 2014). Some of these networks distribute parts of their historical data via the International Soil Moisture Monitoring Network (https://ismn.earth/en/networks, Dorigo et al., 2021). In Europe, CRNS data from individual institutes were consolidated by Bogena et al. (2022).

For Germany, a nationally coordinated effort is not yet in place, although some locations have been instrumented as part of the TERENO observatories (Zacharias et al., 2024). Furthermore, CRNS is being used to monitor selected cropland sites in parts of the federal state of North Rhine-Westphalia (Ney et al., 2021).

For the federal state of Brandenburg, a consortium of research institutions and state agencies was formed in 2024 in order to implement and maintain a CRNS-based soil moisture monitoring network, designed to represent typical combinations of land use, soil and groundwater conditions in the state.

One aim of this paper is to introduce this monitoring network to the community, including its design, its evaluation, and the results of more than one year of operation. Furthermore, due to the high vertical and horizontal representativeness of the CRNS-based soil moisture observations, the statewide network deployment provides a new opportunity - a new reference - to assess the validity of soil moisture products for the Brandenburg region - let it be from modelling or remote sensing. That way, we could leverage the observational records to expand the limited temporal, vertical and horizontal coverage of the mere instrumental monitoring, i.e. to use the observational data in order to develop and assess soil moisture products specifically for the state of Brandenburg. This will be exemplified in a case study that is guided by three questions: (1) How do widely used large-scale soil moisture products (such as ERA5-Land or the Soil Water Index of the Copernicus Land Monitoring Service, CLMS) capture the observed soil moisture dynamics in comparison to a local soil hydrological model (Soil-Water-Atmosphere-Plant, SWAP) that was set up on the basis of region-specific data? (2) Can the CRNS-based soil moisture estimates help to improve such products, e.g., by means of bias correction? (3) What are the implications of this evaluation for the application of such products in the management of water-related risks in the state of Brandenburg? Which products show the best prospects, depending on the application context?

In section 2, we will present typical characteristics of state of Brandenburg (2.1), introduce the CRNS-based soil moisture monitoring network (2.2), the estimation of soil moisture from neutron data (2.3), the local soil hydrological model setup (2.4.1), the selected large-scale soil moisture products from modelling and remote sensing (2.4.2), and the approach to benchmark soil moisture products in reference to the soil moisture estimates from our observational network (2.5). In section 3, we will present and discuss the soil moisture estimates from the first year of network operation (3.1), the evaluation of soil moisture products from modelling and remote sensing (3.2), and discuss resulting implications for the management of water-related risk in the state of Brandenburg (3.3). We will also discuss some practical lessons learned from the first year of operation (3.4). Section 4 will conclude, and provide an outlook on prospective research and applications that could emerge from the presented monitoring effort.

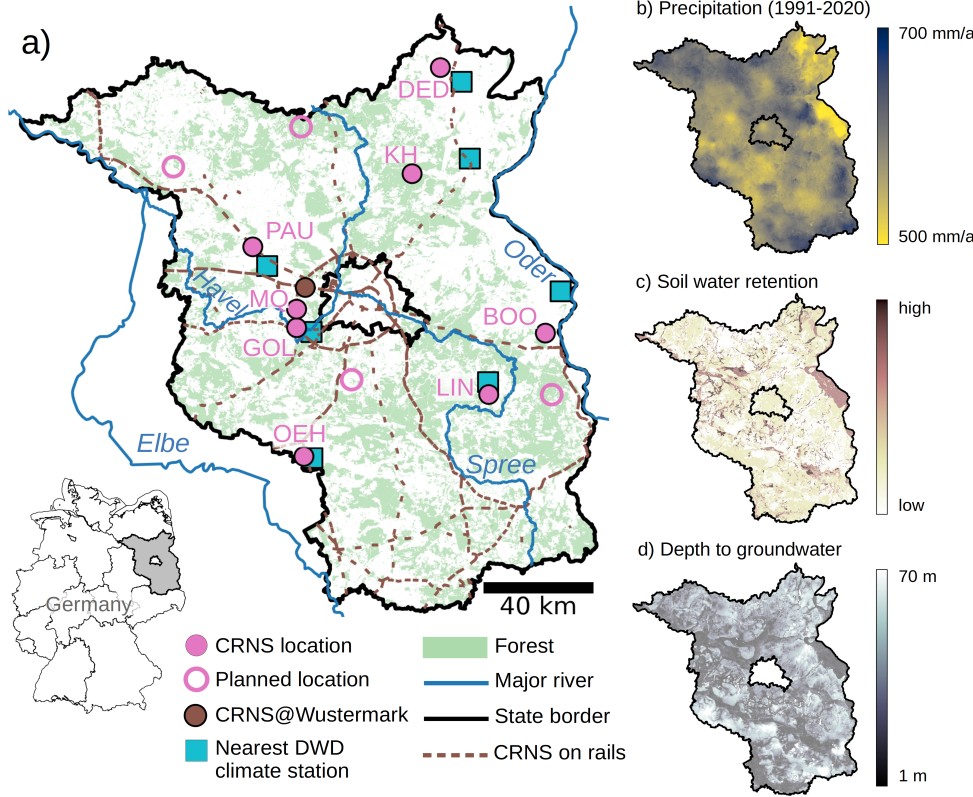

**Figure 1.** (a) Current locations of CRNS-based soil moisture monitoring network (filled pink dots), planned locations (hollow pink dots) nearest DWD climate stations (turquoise squares), forest coverage (green shade, © OpenStreetMap contributors, 2024, distribution under ODbL license), and rail-based CRNS network (brown). The labels show the location identifiers in reference to Tab. 1. (b–d) Spatial distribution of other geographical attributes across Brandenburg: mean annual precipitation (1991-2020, based on DWD's HYRAS-PRE data), top soil water retention capacity (LBGR, 2024), and depth to the groundwater table (LfU, 2013).

## 2 Data and methods

### 2.1 Study area

The federal state of Brandenburg (Fig. 1), with an area of 29,479 km$^2$, has a temperate continental climate with cold winters, warm summers, and moderate precipitation that is evenly distributed throughout the year (mostly Cfb climate according to the classification of Koeppen and Geiger). Conditions are slightly drier in the east while maximum annual precipitation occurs in the north west and in the south. 40% of the state are characterized by shallow groundwater tables (LfU, 2013) of $\leq 3$ m below the surface (lowland areas, mostly Urstromtäler from the Weichsel glacial period); the groundwater table depth is at 3–15 m in 38% and >15 m in 22% of the area (below the elevated areas, often consisting of moraines from the Weichsel epoch, except the south and west which are dominated by the Saale epoch). Soils are dominated by highly permeable sandy soils and loamy

sands that together account for 85% of the state, as well as organic soils (8%) in the very wet lowlands (LBGR, 2024). The land use in Brandenburg (Amt für Statistik Berlin-Brandenburg, 2023) is composed of 36% cropland, 11% agriculturally used grassland, 37% forests (most frequent species: 69% Scotts pine, Pinus sylvestris, 8% common oak, Quercus robur, and 4% common beech, Fagus sylvatica, MLUK, 2024), 3% of other vegetated areas, 10% settlements and traffic infrastructure, and 4% surface waters.

## 2.2 The monitoring network

In a transdisciplinary effort, five institutions have combined their resources to establish a CRNS-based network for soil moisture in Brandenburg: the Helmholtz Centre for Environmental Research (UFZ), the University of Potsdam (UP), the Ministry of Economy, Labor, Energy and Climate Protection (MWAEK), the State Agency for Mining, Geology and Resources (LBGR), and the State Environment Agency (LfU). Other institutions (namely the Leibniz Centre for Agricultural Landscape Research, and the State Forestry Agency) as well as private land owners contributed by providing the permission to use suitable monitoring sites. The network is designed as a long-term monitoring effort, facilitated by the close collaboration between state agencies and research institutions in which the latter took the initial lead in instrumentation, maintenance, data processing, and dissemination, while the state agencies are progressively assuming such responsibilities.

By June 2024, eight locations (see Fig. 1 and Tab. 1) were equipped with a CRNS station. Such a station includes a neutron detector, logger and telemetry, solar power supply, and sensors for barometric pressure, temperature, and humidity, as well as conventional point-scale sensors (SMT100, manufactured by Truebner) of soil moisture in various measurement depths as an additional reference. Five of these stations were instrumented in the first half of 2024, three had already been operational before (Lindenberg since 2020, Marquardt and Oehna since 2022). Four additional locations are expected to be instrumented by November 2025. For the selection of sites, various criteria had to be accounted for:

- The monitoring locations should represent the major landscape attributes of Brandenburg that are considered to govern soil water dynamics and the surface water balance, namely climate, land use, soil texture and depth to the groundwater table. According to the location attributes presented in Tab. 1 and the prevalence of these attributes in Brandenburg as outlined in section 2.1, the selected locations account for 84% of the state's area in terms of land use (cropland, grassland, forests, namely pine, oak and beech forests), 81% in terms of soil texture classes (medium sand, loamy sand, fine sand), and 86% in terms of groundwater table depth. While the network in its current form does not yet sufficiently cover the south of Brandenburg, the spatial distribution of monitoring locations accounts for a large part of the (moderate) spatial variability of the climate, particularly precipitation.

- Accessibility and permission of the land owner to place and maintain a CRNS station was mandatory.

- Availability of security measures against theft or vandalism, typically by fences and limited visibility of the equipment from roads and other public places, was mandatory.

- Sufficient network coverage for remote data transmission was mandatory.

- Sensor footprints that are homogeneous with regard to these attributes were preferable because of a better interpretability of the CRNS signal.

- Existing additional instrumentation (e.g., by groundwater level sensors, lysimeters, climate gauges, complementary soil moisture measurements) was not mandatory, but advantageous.

- Another optional criterion was the relative proximity to railway tracks. A pilot study in the Harz mountains in central Germany has recently demonstrated that rail-based CRNS can monitor soil moisture along landscape transects of several kilometres at daily resolution (Altdorff et al., 2023). Building on this concept, four additional rail-based CRNS systems are now operational in Brandenburg (Fig. 1) and adjacent federal states. Almost each day, these systems collect soil moisture data along hundreds of kilometres (although the routes vary, depending on the operational schedule), and

transmit the data in near real-time. Although this approach is still in its early stage, the placement of monitoring locations in the vicinity of railway tracks will enable future comparisons of measurements by overlapping sensor footprints.

**Table 1.** Overview of CRNS-based monitoring locations. The CRNS systems "StyX S2" were manufactured by StyX Neutronica GmbH (Mannheim, Germany), "CRS-1000", "CRS-2000" and "CRS-2000-B" by Hydroinnova LLC (Albuquerque, USA). Soil texture classes were obtained from BUEK300 (LBGR, 2024): Sl2 (loamy sand), fSms (fine sand), mSfs (medium sand); depth to the groundwater table was obtained from LfU (2013). "Additional instrumentation" indicates the availability of hydrometeorological or hydrological monitoring by the site owners (such as eddy flux towers, lysimeter, groundwater observation wells) in addition to the CRNS station itself.

| ID | Location name | CRNS system | Land use | Soil texture | Depth to groundwater (m) | Additional instrumentation | Nearby rail track |
|---|---|---|---|---|---|---|---|
| *Instrumented by June 2024* | | | | | | | |
| BOO | Booßen | StyX S2 | Cropland | Sl2 | 10 | No | No |
| GOL | Golm | StyX S2 | Grassland | fSms | 2 | No | Yes |
| PAU | Paulinenaue | StyX S2 | Grassland | fSms | 2 | Yes | Yes |
| DED | Dedelow | StyX S2 | Cropland | Sl2 | 10 | Yes | No |
| KH | Kienhorst | StyX S2 | Pine forest | mSfs | 7.5 | Yes | No |
| MQ[1] | Marquardt | CRS-1000 | Cropland | Sl2 | 15 | Yes | Yes |
| LIN | Lindenberg | CRS-2000 | Grassland | Sl2 | 3 | Yes | No |
| OEH[2] | Oehna | CRS-1000 | Cropland | Sl2 | 15 | No | No |
| *To be instrumented by November 2025* | | | | | | | |
| TRB | Trebbin | CRS-2000-B | Cropland | Sl2 | 15 | No | No |
| FUE | Fünfeichen | CRS-2000-B | Oak forest | mSfs | 20 | Yes | No |
| SHG | Schönhagen | CRS-2000-B | Cropland | mSfs | 10 | No | No |
| BEE | Beerenbusch | CRS-2000-B | Beech forest | mSfs | 20 | Yes | No |

[1] not part of the CRNS cluster described by Heistermann et al. (2023); [2] intermittent pivot irrigation in dry periods;

## 2.3 Retrieval and evaluation of soil moisture estimates from CRNS observations

Cosmic-Ray Neutron Sensing (CRNS) is based on the detection of neutrons generated by the interaction of cosmic radiation with Earth's atmosphere. At ground level, the intensity of these neutrons is inversely related to the abundance of hydrogen in the near-surface environment (Desilets et al., 2010). While soil water typically constitutes the largest hydrogen pool, other occurrences of hydrogen may have to be accounted for (such as in vegetation, soil organic matter, or snow).

We estimate volumetric soil moisture from neutron intensities ($\theta_{\text{CRNS}}$, $m^3\,m^{-3}$) according to a procedure referred to as "general calibration" which is documented in Heistermann et al. (2024) and was validated on an extensive dataset of 75 CRNS stations across Europe, as compiled from various published datasets. We refer to Heistermann et al. (2024) with regard to the details of the method. In essence, the observed neutron intensities are converted to volumetric soil moisture by means of a non-linear transformation function (Desilets et al., 2010), using a uniform ("general") value of 2306 cph for the calibration parameter $N_0$. In order to allow for the application of such a uniform $N_0$, the observed neutron intensities have to be standardised by accounting for a range of effects which we will only briefly summarize here (see Heistermann et al., 2024, for details):

- The **sensitivity of the neutron detector** relative to a known reference is required to standardize neutron count rates to a common level. To obtain the relative sensitivity, each sensor of the network was collocated to a sensor of known sensitivity for at least two days. The resulting relative sensitivity factors $f_s$ are presented in Tab. 4 in section 3.1.

- The spatial variation of **incoming cosmic radiation** was accounted for by using the PARMA model (Sato, 2015) while the temporal variation was corrected for by using time series of the neutron monitor on the Jungfraujoch ("JUNG" in the neutron monitor database, https://www.nmdb.eu/nest).

- For eliminating the temporal **effects of barometric pressure and atmospheric humidity**, we used time series that were recorded locally at each CRNS station.

- **Soil organic carbon** (SOC) and **lattice water** (LW) content ($kg\,kg^{-1}$) as well **soil dry bulk density** ($\rho_b$, $kg\,m^{-3}$) in the sensor footprint were obtained during soil sampling campaigns: at a minimum of four randomly chosen locations within the near range (20 m radius) of the sensor, cylinder samples were extracted from the upper 30 cm of the soil at increments of 5 cm. For obtaining average values of these variables for the sensor footprint, we followed the weighting procedure outlined by Schrön et al. (2017). Sampling within the inner 20 m of the footprint constituted a trade-off between available workforce and representativeness. Resulting uncertainties will be discussed in section 3.1.

- Heistermann et al. (2024) showed that the effect of **biomass** on the uncertainty of CRNS-based soil moisture estimates is negligible for grassland and cropland sites. For such sites (see Tab. 1), dry above-ground biomass (AGB) density was set to a constant value of $1\,kg\,m^{-2}$. For the forest site (Kienhorst), the average dry AGB density was determined to a value of $11\,kg\,m^{-2}$, based on allometric relationships together with extensive measurements of the breast height diameter at 33 specimen of Pinus sylvestris at the Level II monitoring plots of the UNECE Convention on Long-range Transboundary Air Pollution (ICP Forests) to which the Kienhorst site belongs.

To evaluate the CRNS-based soil moisture estimates ($\theta_{CRNS}$), reference observations within each CRNS footprint were ob-
tained from the aforementioned soil sampling campaigns: Following Fersch et al. (2020), first, volumetric soil moisture was
obtained for each of the four profiles sampled with cylinders; second, soil moisture profiles (30 cm depth, 5 cm increments) were
measured by impedance-based soil moisture sensors (ThetaProbe ML2x, Delta-T Devices LLC, Cambridge, UK) at a minimum
of 18 additional locations per footprint. The impedance-based measurements were calibrated to the collocated cylinder-based
measurements. The cylinder- and impedance-based measurements were then averaged vertically and horizontally by using the
weighting functions established by Schrön et al. (2017), resulting in an average value of $\theta_{REF}$ that was considered as the refer-
ence value representative for the CRNS footprint. For the weighting procedure, we first interpolated the profile measurements
to a resolution of 1 cm. Since the vertical and horizontal weights depend on the average soil moisture itself, this average was
retrieved iteratively, as suggested by Schrön et al. (2017): starting with an initial guess of soil moisture (obtained from the
arithmetic mean of measurements from all profiles), each iterative step consisted of (1) computing the vertically weighted
average per profile, (2) computing the horizontally weighted average per footprint. As the weights also depend on the dry bulk
density, the corresponding footprint average of dry bulk density was computed simultaneously in the same iterative procedure
from the available measurements (see above). The iteration was interrupted after the estimates did not change by more than
0.1% (typically the case after three to four iterative steps). The resulting value of $\theta_{REF}$ was then used to assess the performance
of the aforementioned estimation procedure by computing the error (difference between $\theta_{CRNS}$ and $\theta_{REF}$) at each location, and
then computing the mean error (ME) and the root mean squared error (RMSE) across all eight monitoring location.

## 2.4 Models and soil moisture products

One of the questions behind this study is how the performance of widely used large-scale soil moisture products compares to a
local soil hydrological model that was set up on the basis of region-specific data. More generally, we intend to illustrate how the
CRNS-based soil moisture estimates ($\theta_{CRNS}$) support the assessment of soil moisture products for the management of water-
related risks in the state of Brandenburg. In this section, we present the soil moisture products that were evaluated against
$\theta_{CRNS}$: first, the set-up and application of the local soil hydrological model, and, second, selected large-scale soil moisture
products from modelling and remote sensing.

### 2.4.1 Local soil hydrological model

We employed the 1-dimensional Soil-Water-Atmosphere-Plant model (SWAP, van Dam et al., 2008) to simulate soil water
dynamics and water fluxes at each monitoring location. SWAP calculates vertical soil water movement by solving the Richards
equation, and hence accounts for infiltration and capillary rise on the basis of soil hydraulic properties and governing boundary
conditions. Evapotranspiration is estimated using the Penman-Monteith equation, considering factors such as soil moisture
content, vegetation type, and atmospheric conditions. This dual focus on soil hydrology and atmospheric interactions allows
for a detailed analysis of the surface water balance and the movement of water through the unsaturated zone towards or from
the groundwater table which is, in our model setup, considered as static (see Tab. 1) and implemented as a Dirichlet boundary
condition. The temporal resolution of the model is one day. The vertical resolution is 1 cm (between a depth of 0-5 cm), 2.5 cm

(at 5-15 cm), then 5 cm (at 15-50 cm), 10 cm (at 50-100 cm), 10 cm (at 100-200 cm), 20 cm (at 200-500 cm) and 50 cm below 500 cm. The actual depth of the soil column depends on the depth of the groundwater surface at the corresponding location (see Tab. 1).

**Table 2.** Overview of key SWAP model parameters related to vegetation and corresponding references. Please refer to Kroes et al. (2017) for further details on the corresponding parameters.

| Parameter name | Meaning | Forest | Grass/cropland | References |
|---|---|---|---|---|
| **Leaves and roots** | | | | |
| GCTB | Max. leaf area index, LAI (-) | 3.5 | 3.0 | LFB (2025), Kroes et al. (2017) |
| RDTB | Rooting depth (cm) | 150 | 40 | Guerrero-Ramírez et al. (2021) |
| **Evapotranspiration** | | | | |
| RSC | Minimum canopy resistance (s/m) | 180 | 130 | Guan and Wilson (2009) |
| **Interception acc. to...** | | | | |
| **...Von Hoyningen-Huene (1983)** | | | | |
| COFAB | Interception coefficient (cm) | – | 0.25 | Kroes et al. (2017) |
| **...Gash et al. (1995)** | | | | |
| PFREE | Free throughfall coefficient (–) | 0.32 | - | Russ et al. (2016) |
| PSTEM | Stem flow coefficient (–) | 0.02 | – | |
| SCANOPY | Storage capacity of canopy (cm) | 0.08 | - | |
| AVPREC | Avg. rainfall intensity (cm/d) | 3.30 | – | |
| AVEVAP | Avg. evaporation int. during rain (cm/d) | 0.46 | – | |

As atmospheric forcing, we used daily climate observations of the German Meteorological Service (Deutscher Wetterdienst, DWD henceforth) at the nearest climate station (Fig. 1) for the following daily variables: minimum and maximum air temperature (°C), average relative air humidity (%), sunshine hours (h), and average wind speed ($\mathrm{m\,s^{-1}}$). For precipitation, we applied DWD's radar-based quantitative precipitation product RADOLAN (DWD, 2022) in order to better capture small-scale convective rainfall at the monitoring locations especially during the summer season. Tab. 2 highlights important vegetation-related model parameters that were used for our study, including the corresponding literature references.

Finally, soil hydraulic parameters (SHP) had to be set in order to represent the relationship between matric potential ($\psi$, hPa) and volumetric soil water content (SWC, $\mathrm{m^3/m^3}$) as well as hydraulic conductivity ($K_s$, $\mathrm{cm\,d^{-1}}$). Using the model of van Genuchten and Mualem (van Genuchten, 1980), the SHP correspond to five parameters: residual water content ($\theta_r$, $\mathrm{m^3\,m^{-3}}$), saturated water content ($\theta_s$, $\mathrm{m^3\,m^{-3}}$), air entry point ($\alpha$, $\mathrm{cm^{-1}}$), the shape parameter of the retention curve (n, dimensionless), and the saturated hydraulic conductivity $K_s$ ($\mathrm{cm\,d^{-1}}$). To obtain SHP values at the monitoring locations, we applied the widely used pedotansfer function ROSETTA (Schaap et al., 2001). As input, ROSETTA requires the fractions of sand, silt and clay, which were obtained from the texture attribute of the state's soil map BUEK300 (LBGR, 2024, see also Tab. 1). This soil map, however, only represents a qualitative soil texture class which, in turn, implies typical ranges of sand ($S$), silt ($Si$) and clay ($C$)

content according to BGR (2005). For the uncalibrated model, we fixed $C$ to a value of 5 percent and set $S$ as the midpoint of the range of sand content as specified by the soil map. $Si$ then equalled the remainder to 100%. Tab. 3 shows the values for $S$, $Si$ and $C$ for each of the three soil texture classes considered in our study, as well as the corresponding SHP values. Please see section 2.5 on how the model was calibrated for each monitoring location.

To allow for the comparison to $\theta_{\mathrm{CRNS}}$, the simulated vertical soil moisture profile at each daily time step was vertically weighted using the weighting function introduced by Schrön et al. (2017).

**Table 3.** Ranges of sand ($S$), silt ($Si$) and clay ($C$) for the three major soil texture classes in Brandenburg (Sl2, mSfs, and fSms) as obtained from LBGR (2024) and BGR (2005); resulting $S$-$Si$-$C$ combinations used for the uncalibrated model, as well as the resulting sets of SHP parameters; corresponding monitoring locations.

| Texture class | Ranges of S-Si-C | S-Si-C set | $\theta_r$ | $\theta_s$ | $\alpha$ | n | $K_s$ | locations |
|---|---|---|---|---|---|---|---|---|
| Sl2 | 67-85, 10-25, 5-8 | 77,18,5 | 0.05 | 0.39 | 0.020 | 1.68 | 131 | BOO, DED, MQ, LIN, QEH |
| mSfs | 85-100, 0-10, 0-5 | 90,5,5 | 0.05 | 0.40 | 0.027 | 2.28 | 441 | KH |
| fSms | 65-75, 20-35, 0-5 | 70,25,5 | 0.05 | 0.39 | 0.017 | 1.58 | 88 | GOL, PAU |

### 2.4.2 Large-scale soil moisture products

For the present case study, we selected three different, widely used products:

– Volumetric soil moisture from **ERA5-Land** was obtained from two soil layers (0-7 cm and 7-28 cm) of the ERA5-Land reanalysis (hourly data, spatial resolution $\approx 9$ km, see Muñoz Sabater, 2025). A vertically weighted mean of these two layers was obtained by using the aforementioned vertical weighting function Schrön et al. (2017). ERA5-Land is available from 1950 to present with a latency of approximately five days, and was also recommended by Zheng et al. (2024) for rather humid climates, based on a comparison with a large set of CRNS observations.

– The **Soil Water Index (SWI)** is generated by applying exponential filtering to a surface soil moisture product (retrieved from ASCAT and Sentinel-1, see Copernicus Land Monitoring Service (CLMS), 2025). The soil depth for which SWI is representative depends on the characteristic time length (T) of the exponential filter. The SWI product provides volumetric soil moisture for T values of 2, 5, 10, 15, 20, 40, 60, and 100 (daily, spatial resolution 1 km, available from 2015 until present). As the soil depth to which a T value corresponds also depends on other soil properties, Raml et al. (2025) recommend "selecting the best matching data" from the different T values. We followed this recommendation by selecting a T value that maximised the correlation with the CRNS-based soil moisture observations within the year 2024 (while data from 2025 was reserved for validation). It turned out that T values of 10, 15 and 20 have the highest correlation; differences, however, were marginal, so we used the SWI with a T value of 15 for all locations.

– The surface soil moisture product provided by **Copernicus Climate Change Service (C3S)** refers to the upper 2–5 cm of the soil (daily, spatial resolution of $\approx 14 \times 22$ km, available from 1978 until present). It is retrieved from a large

set of spaceborne sensors and represents the current state-of-the-art for satellite-based soil moisture climate data record production (Copernicus Climate Change Service and Copernicus Climate Change Service, 2018). The data includes three products (active and passive, and combined). In order to simplify the analysis, we selected the "combined" product which quantifies volumetric surface soil moisture and exhibited, in comparison to active and passive alone, the highest correlation with the CRNS-based soil moisture observations in the year 2024 (while again data from 2025 was reserved for validation).

For all products, the soil moisture time series at the monitoring locations were obtained by choosing the nearest grid cell.

## 2.5 Bias correction, model calibration, and performance evaluation

As we will see in section 3.2, some soil moisture products suffer from high levels of systematic bias. We investigated the potential to remove the bias at least locally, and thereby increase the usefulness of these products for specific application scenarios (see section 3.3). For the large scale products SWI, ERA5-Land and C3S (see section 2.4.2), a simple multiplicative local bias correction was applied: using only data from 2024, we computed, at each monitoring location, the ratio between the mean value of the observed soil moisture ($\theta_{\mathrm{CRNS}}$) and the mean value of the corresponding soil moisture product and multiplicatively applied this factor to the entire time series of the product at this location, under the (admittedly strong) assumption that the bias is constant over time (the validity of which was tested on the data from 2025).

For the local soil hydrological model SWAP, we chose a different approach. On the one hand, a bias correction was desirable also for this model in order to allow for a fair comparison to the bias-corrected large-scale models. On the other hand, we intended to maintain the model's consistency with the state's soil map, and also to ensure that the simulated values of soil moisture and water fluxes remained physically consistent. In order to meet both criteria, we applied a procedure that could be framed as a "fine tuning of the local sand content". It accounts for the fact that actual soil texture values could vary across locations even in case they have the same texture class. We hence "calibrated" the sand content of each location, but only within the ranges specified by the corresponding texture class (see Tab. 3, still fixing $C$ to a value of 5 percent and treating $Si$ as the remainder to 100%). To that end, the sand content $S$ was set to a value that minimised the mean absolute difference between simulated soil moisture and CRNS-based soil moisture. This was carried out by exclusively using data from the year 2024 while data from 2025 was reserved for validation. While the results of the validation are presented in section 3.2, the calibration results of the SWAP model (resulting texture values and SHP sets, performance metrics over calibration period) are shown in the supplementary material (Tab. S1).

For an independent evaluation of each product – with and without bias correction (or calibration) – we used the soil moisture observations ($\theta_{\mathrm{CRNS}}$) from January 1 to September 1, 2025. For each location, we computed the root mean squared error (RMSE), the percent bias (PBIAS), the Pearson correlation coefficient (r), and the Nash-Sutcliffe Efficiency (NSE). For the products without bias-correction, we additionally pooled the observations and predictions across all monitoring locations, and computed the aforementioned metrics for this pooled dataset. This approach served to evaluate how the products performed in capturing the variability of soil moisture across monitoring locations (spatial variability) which is important to assess the

potential for spatial upscaling of soil moisture estimates beyond the limited set of monitoring locations. This analysis was not carried out for the bias-corrected products, since the local bias correction factors are not directly transferable in space.

## 3 Results and discussion

### 3.1 CRNS-based soil moisture estimation

As pointed out in section 2.3, we applied the so-called general calibration function (Heistermann et al., 2024) in order to estimate the volumetric soil water content $\theta_{\text{CRNS}}$ from observed neutron count rates. The main motivation behind this approach was to avoid point measurements of SWC as a source of uncertainty for the local calibration of the conversion function. To apply the general calibration function, however, we required the sensitivity of the neutron detector relative to a reference detector ($f_s$), and several site-specific variables such as the gravimetric soil water equivalents of soil organic carbon and lattice water ($\theta_{SOM}^g$

and $\theta_{LW}^g$), dry aboveground biomass (AGB) density, and soil dry bulk density ($\rho_b$). Based on the data collection outlined in section 2.3, Tab. 4 reports the site specific values of these parameters, as well as $\theta_{\text{REF}}$ obtained from manual sampling at the date of the sampling campaign, and the corresponding value of $\theta_{\text{CRNS}}$.

**Table 4.** Parameters for CRNS-based SWC estimation (see main text), $\theta_{\text{CRNS}}$ and $\theta_{\text{REF}}$ at the sampling dates and the corresponding difference $\theta_{\text{CRNS}} - \theta_{\text{REF}}$ between these values.

| ID | $f_s^{-1}$ | $\theta_{SOM}^g + \theta_{LW}^g$ | $\rho_b$ | Sampling date | $\theta_{\text{CRNS}}$ | $\theta_{\text{REF}}$ | $\theta_{\text{CRNS}} - \theta_{\text{REF}}$ |
|---|---|---|---|---|---|---|---|
| | (–) | (kg/kg) | (kg m$^{-3}$) | | (m$^3$ m$^{-3}$) | (m$^3$ m$^{-3}$) | (m$^3$ m$^{-3}$) |
| BOO | 1.20 | 0.016 | 1420 | 2024-09-04 | 0.020 | 0.069 | −0.049 |
| GOL | 1.13 | 0.045 | 1080 | 2024-09-03 | 0.084 | 0.120 | −0.036 |
| PAU | 1.15 | 0.060 | 1510 | 2024-09-19 | 0.142 | 0.190 | −0.048 |
| DED | 1.06 | 0.018 | 1470 | 2024-10-16 | 0.155 | 0.200 | −0.045 |
| KH | 1.13 | 0.017 | 1030 | 2024-09-18 | 0.050 | 0.076 | −0.026 |
| MQ | 0.49 | 0.015 | 1280 | 2023-05-17 | 0.132 | 0.097 | 0.035 |
| LIN | 0.87 | 0.014 | 1430 | 2021-11-19 | 0.165 | 0.199 | −0.034 |
| OEH | 0.45 | 0.009 | 1500 | 2024-04-05 | 0.149 | 0.148 | 0.001 |

**RMSE = 0.037**

**ME = −0.025**

As we did not use $\theta_{\text{REF}}$ for local calibration, we could use it to assess the uncertainty of the general calibration procedure, albeit being aware that more than one value of $\theta_{\text{REF}}$ per footprint would be preferable for a comprehensive assessment, and that

$\theta_{\text{REF}}$ itself is also subject to considerable uncertainty due to the high small-scale variability of soil moisture in combination with the limited representativeness of the point measurements (still, the number of sampling profiles to obtain $\theta_{\text{REF}}$ is quite high, with at least four profiles with cylinder samples and at least 18 profiles with impedance-based measurements in each sensor footprint, see section 2.3). Overall, the absolute difference between $\theta_{\text{CRNS}}$ and $\theta_{\text{REF}}$ was always lower than 0.05 m$^3$ m$^{-3}$. The

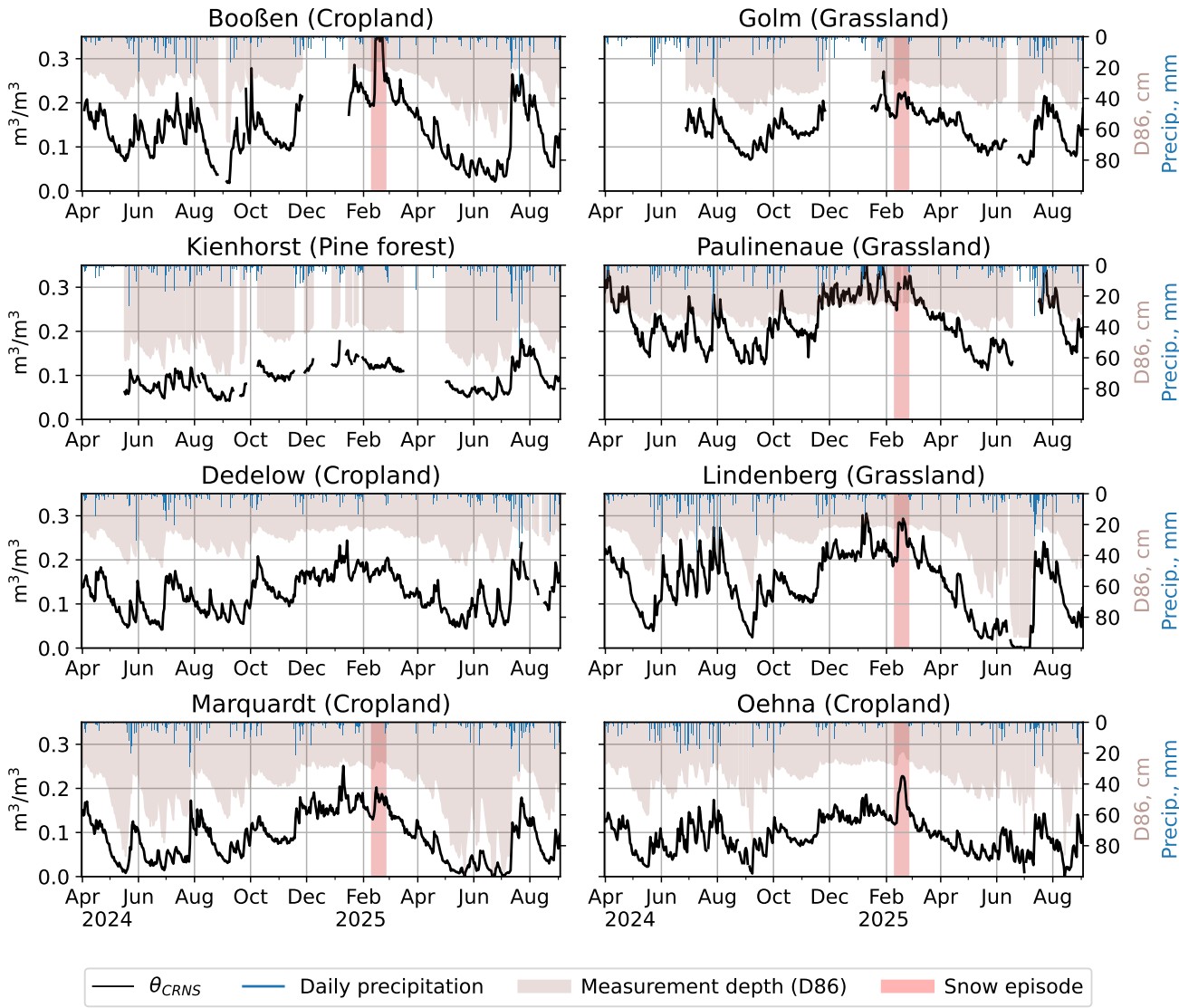

**Figure 2.** Observed (black) soil water dynamics at the monitoring locations, as well as CRNS measurement depth (D86, i.e. the depth that accounts for 86 % of the signal) and daily precipitation depths. Extended snow episodes are marked by the red shade. During these times, the CRNS signal should not be interpreted in terms of soil moisture.

root mean squared error (RMSE) amounted to $0.037\,\mathrm{m^3\,m^{-3}}$ which is, in our view, a satisfactory agreement given the absence of any local calibration. As the expensive ground-based reference measurements are commonly all used for the local CRNS calibration, only few studies are available which carried out such an independent validation of the CRNS-based soil moisture estimates, either based on additional sampling campaigns or based on continuously measuring sensor networks: for instance, Cooper et al. (2021) stated that "repeat calibrations using secondary samples have been conducted at two COSMOS-UK sites to

explore the accuracy of the derived VWC obtained on a particular day [...]" and that "there was below $0.03\,\mathrm{m^3\,m^{-3}}$ difference in volumetric water content". Coopersmith et al. (2014) used an in-situ network at one COSMOS station for validation and found the RMSE "well below $0.04\,\mathrm{m^3\,m^{-3}}$". Schrön et al. (2017) followed a similar approach and found RMSE values between $0.006$ and $0.051\,\mathrm{m^3\,m^{-3}}$ across four CRNS sites in Germany, three of which belong to the TERENO program. Finally, Iwema et al. (2015) systematically investigated the effect of the number of calibration measurements at two TERENO sites in Germany and found mean absolute errors between about $0.04$ and $0.07\,\mathrm{m^3\,m^{-3}}$ in the validation (depending on the number of calibration dates from one to six). Altogether, these references are quite in line with the RMSE obtained for the eight CRNS stations in Brandenburg. Still, care needs to be taken when comparing such metrics across different environmental conditions (namely across different wetness regimes).

The mean error (ME) of $-0.025\,\mathrm{m^3\,m^{-3}}$, however, indicates that a large portion of the RMSE is due to a systematic underestimation of the soil moisture by $\theta_{\mathrm{CRNS}}$. This result is in line with some recent studies, including Heistermann et al. (2024), which suggest the use of a new type of conversion function recently published by Köhli et al. (2021). The original functional form suggested by Desilets et al. (2010) and also adopted by Heistermann et al. (2024) tends to underestimate soil moisture under very dry conditions. For future applications, we hence recommend to systematically assess the function from Köhli et al. (2021) for CRNS-based soil moisture estimation in Brandenburg. Another major source of uncertainty could be the limited number of four sampling points for obtaining average values of soil organic carbon as well as soil dry bulk density in the sensor footprint. However, these uncertainties are not assumed to introduce any systematic bias. The uncertainty from the aboveground biomass estimation is relatively low for the grassland and cropland sites (Heistermann et al., 2024). For the forest site, the uncertainty of the biomass estimate is potentially higher, yet, in this case the estimate is based on a considerable number of measurements of breast height diameter (see section 2.3). The uncertainty introduced by other parameters of the general calibration approach are considered as relatively low in the context of the present study (specifically, a lot of effort was taken to determine the relative sensitivity of the neutron detectors). It is, however, difficult to explicitly disentangle the different sources of uncertainty.

Fig. 2 illustrates the observed soil water dynamics from April 2024 (when the majority of CRNS stations was operational) until September 2025, together with the measurement depth. The latter accounts for the dynamic effect of soil moisture on the neutron signal and was obtained by applying the vertical weighting function from Schrön et al. (2017) in order to obtain the depth that accounts for 86% of the neutron signal (D86, see Schrön et al., 2017). Based on this figure, we can maintain the following:

– The observed seasonal dynamics are consistent and plausible across monitoring locations, but also illustrate how both the short term behaviour (as a results of different precipitation dynamics) as well as the average moisture levels (as a result of different site characteristics) differ between locations, underpinning the usefulness of the monitoring.

– For some sites (specifically Lindenberg and Marquardt), the soil moisture approaches values close to zero during very dry periods. While residual water contents below $0.05\,\mathrm{m^3 m^{-3}}$ are not uncommon for sandy soils (see, e.g., Vereecken et al.,

2007), we hypothesize, based on the aforementioned underestimation, that the application of the conversion function recently published by Köhli et al. (2021) could mitigate the issue of overly low soil moisture estimates.

- The measurement depth varies considerably in time and space (across locations), as a result of soil moisture variability (typically between around 25–30 cm in the wet season, and 40–60 cm in the dry season, with some higher values for extremely dry conditions, see previous point).

- For the locations that started before or in April 2024, the records demonstrate clear differences between 2024 and 2025 with regard to spring and early summer. This period, specifically May and June, is critical with regard to the impact of drought on crops (see Brill et al., 2024, for a Brandenburg-specific analysis). While 2024 featured some pronounced drying from mid April to mid May, the following months were characterized by repeated and substantial rainfall, only followed by a dry spell in August 2024. In 2025, however, remarkable drying already started in March and led to a prolonged drought period in May and June (most prominent in Booßen, Lindenberg, and Marquardt), interrupted by a wet July, and followed by another remarkable dry-up in August.

- For six out of eight sites, a pronounced snow episode (at least for conditions in Brandenburg) occurred in February 2025 which well illustrates the fact that during such episodes, the CRNS signal cannot be directly interpreted in terms of soil moisture because of the additional presence of the hydrogen in the snow layer. While this issue should not directly affect the usability of the data for the investigation of drought, we recommend to remove snow-affected periods from the data in case it is used for, e.g., calibration and validation of hydrological models. For that purpose, we recommend snow monitoring data at the DWD climate stations.

- From a technical perspective, the stations in Booßen, Golm, Kienhorst and Paulinenaue were affected by losses of data and resulting gaps which were due to various reasons, including failures of remote data transmission, solar power supply, but also sensor noise that had to be addressed by firmware updates (see also section 3.4).

## 3.2 Evaluation of soil moisture products

Fig. 3 shows the performance metrics of the benchmarked soil moisture products for the independent validation period from January to August 2025.

For the native products (i.e., without bias correction or calibration), all large-scale products (SWI, ERA5-Land, C3C) suffer from very high levels of systematic bias (Fig. 3a) which directly propagates to RMSE and NSE (Fig. 3b and c). The local hydrological model SWAP is much less biased (highest levels for locations MQ and QEH). For RMSE and NSE, SWAP also outperforms all competitors at all locations. For correlation (Fig. 3e), ERA5-Land and SWAP perform similarly, with ERA5-Land even slightly outperforming SWAP in three out of eight locations. Together with its strong bias, the high correlation found for ERA5-Land gives rise to the expectation that this could particularly benefit from a bias correction (see below). In terms of correlation at the individual locations, SWI and C3C are quite similar, with an intermediate performance.

Before discussing the results for bias-corrected products, we would like to highlight the results in the "ALL" column of Fig. 3a-d. The metrics in that column were computed from a dataset that pooled observations and predictions across all locations.

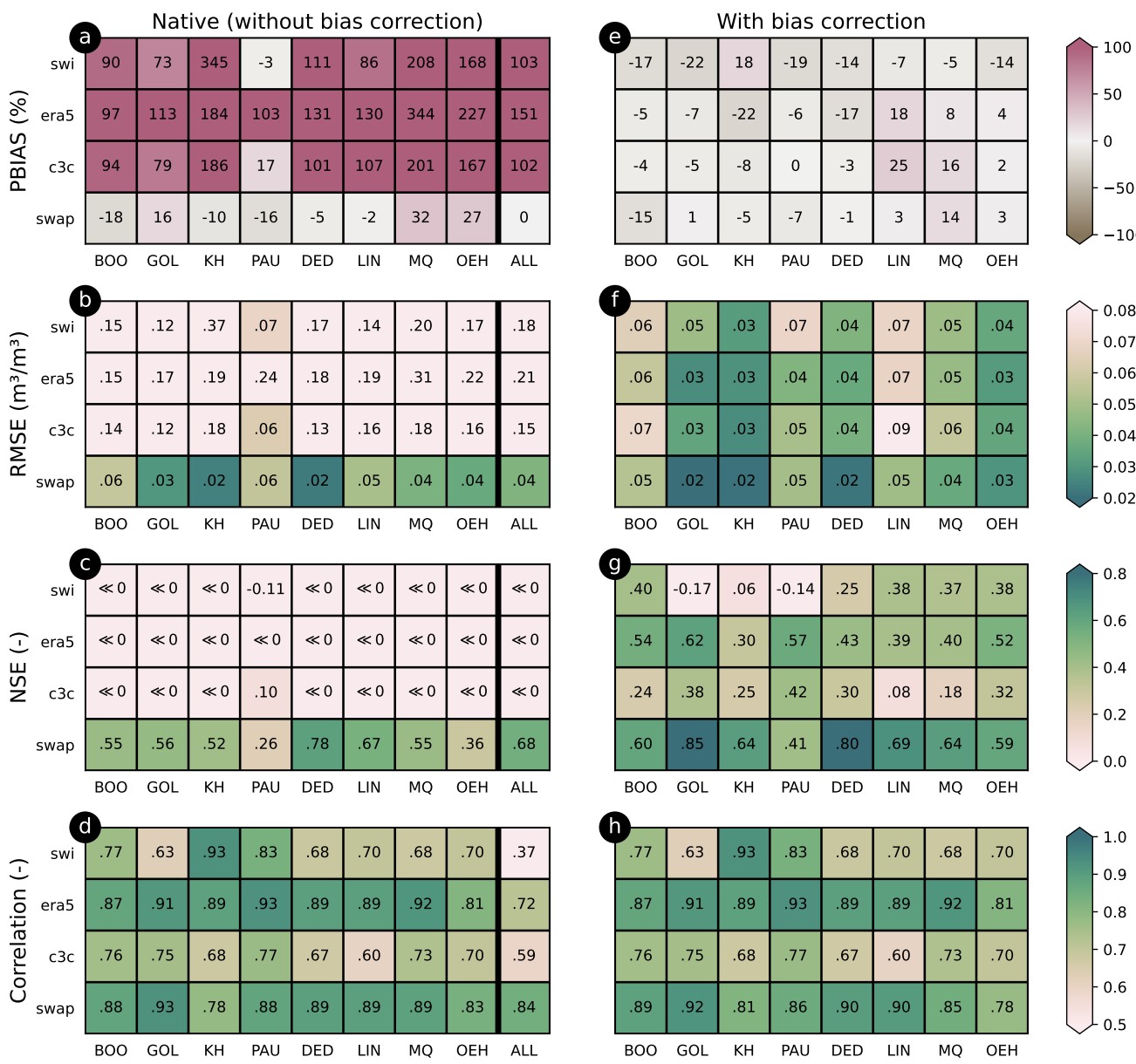

**Figure 3.** Performance metrics (RMSE, PBIAS, Pearson correlation coefficient) for various soil moisture products (swi: CLMS Soil Water Index; era5: ERA5-Land soil moisture; c3c: Copernicus Climate Change Service surface soil moisture, swap: Soil Water Atmosphere Plant model; see section 2.4 for further details) during the validation period from January 1 until August 31, 2025. The text labels within the boxes specify the values of the metrics. Left: native products without bias correction; right: with bias correction.

In that column, SWAP is clearly superior for all metrics, with an NSE of 0.68 which corresponds to "satisfactory" according

to Moriasi et al. (2015). This is particularly important as it highlights the ability of the uncalibrated SWAP model to account for the spatial variability between the monitoring locations, an aspect that is specifically relevant for the prospects of spatial upscaling. Please note that the "ALL" column was not computed for the bias-corrected products: since the bias correction was individually carried out for each location, the results do not hold any information with regard to transferability in space (or, in other words, from one location to the other). Identifying spatially transferable bias correction factors might be possible by taking into account auxiliary environmental variables which was, however, beyond the scope of this study.

Fig. 3e-h illustrate the success of the bias correction (or, for the SWAP model, the calibration of the local sand content). Most PBIAS values (Fig. 3e) range between -20 and 20%, indicating that the assumption of a constant bias is useful to address the massive bias levels present in the native large scale products (Fig. 3a). The bias correction strongly affects RMSE and NSE, confirming the assumption that the poor performance of the native SWI, ERA5-Land and C3C products was largely bias-induced. As already suspected above, ERA5-Land benefits most from the bias correction while the calibrated SWAP model largely outperforms the large scale products in terms of RMSE and NSE (except for location PAU where ERA5-Land is superior). For all locations, the bias-corrected ERA5-Land achieves better ratings in terms of RMSE, NSE and correlation than SWI and C3C. The correlation metric remains, of course, unaffected by the multiplicative bias correction of SWI, ERA5-Land, and C3C; for the SWAP model, it changes only marginally after local calibration of the sand content (Fig. 3h).

To get a better impression of the temporal dynamics behind the performance metrics, Fig. 4 shows the time series of the bias corrected products from April 2024 until September 2025. The figure confirms that the calibrated SWAP model manages best to capture the observed soil moisture dynamics. ERA5-Land also performs quite well, but often struggles to fully represent the seasonal soil moisture amplitudes. This becomes particularly obvious for the Marquardt location in which ERA5-Land overestimates in summer and underestimates in winter. The C3C product appears to be overly noisy. In the scope of the present study, we did not investigate this behaviour in further depth; yet, a higher level of temporal variability is to be expected as C3C is purely satellite-based and only intended to be representative for the upper 2–5 cm. The SWI product shows a similar seasonal behaviour as the ERA5-Land product (partly better, e.g. for Paulinenaue and Dedelow) but generally tends to be too smooth, a property that is subject to the selection of the exponential filter length.

To sum up the evaluation results, the products based exclusively (C3C) or to a large extent (SWI) on satellite-borne soil moisture retrievals do not appear to add much benefit in comparison to the model-based products (ERA5-Land, local SWAP model). The native large-scale products (SWI, ERA5-Land, C3C) suffer from substantial levels of bias (which are heterogeneous across locations). Of all large-scale products, ERA5-Land shows the largest potential to capture the spatial variability of soil moisture across locations (Fig. 3d, column ALL). A simple bias-correction could remove the local bias, however, the resulting bias correction factors are not directly transferable in space. After the bias correction, ERA5-Land is clearly superior to its satellite-based competitors. The local SWAP model mostly outperforms its competitors in terms of PBIAS, RMSE and NSE, with and without bias correction. Of course, it should be clear that these statements are only valid for the selected products. While these are widely used, other products might be available at the national, European or global scale that might show a better performance. For all large-scale products, we also need to keep in mind the spatial mismatch of the gridded products with the horizontal footprint of the CRNS measurement which compromises direct comparability. For an in-depth

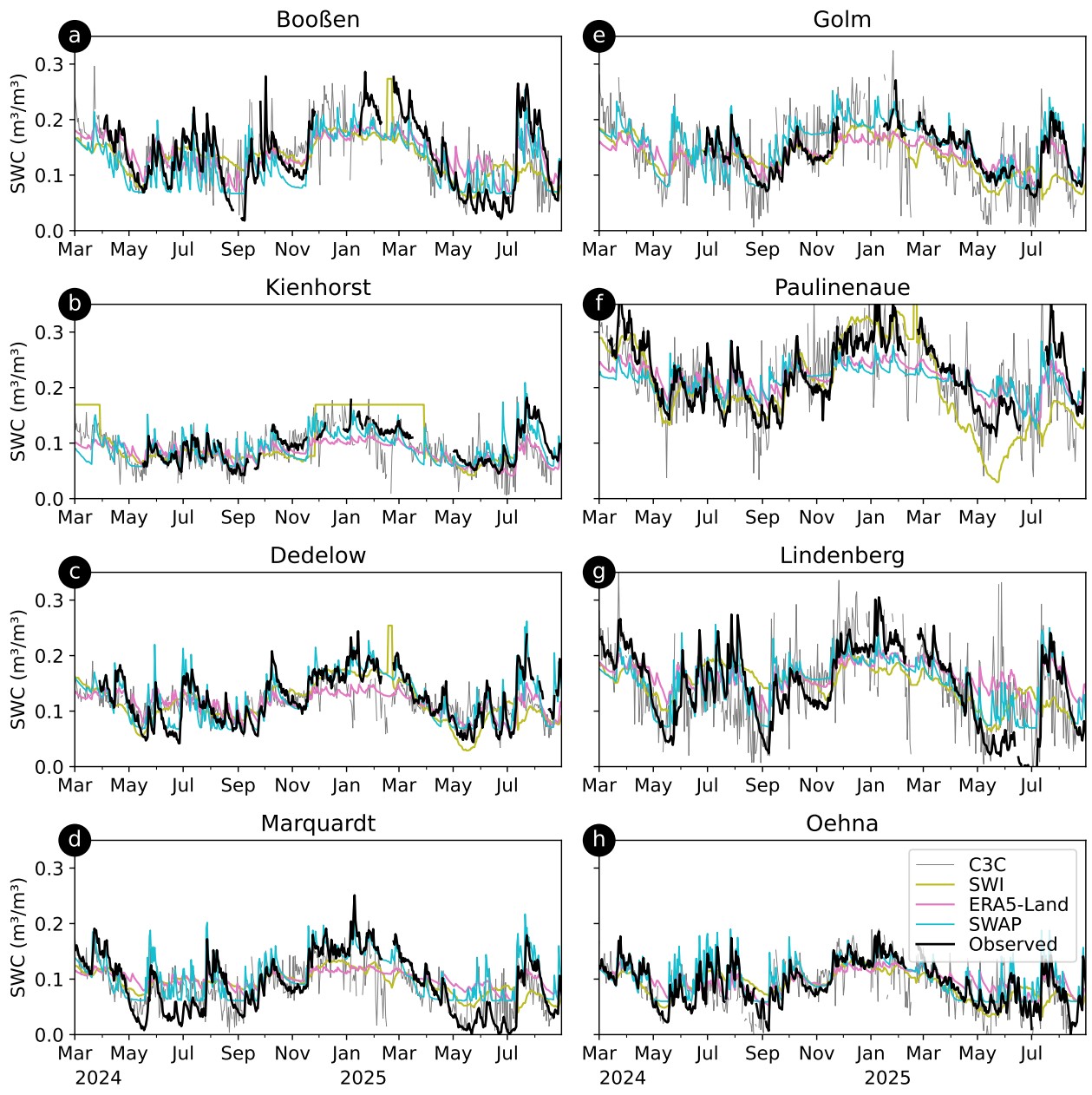

**Figure 4.** Observed soil moisture (black) and bias-corrected soil moisture products at all monitoring locations.

discussion of these issues, especially in the context of comparison to CRNS measurements, we refer to Schmidt et al. (2024) who also address the effects of land cover type, mean annual soil moisture, retrieval algorithm, data quality control, and sensor properties.

In the context of the present study, we maintain that the CRNS-based soil monitoring can serve as a basis to evaluate and improve soil moisture products which in turn have the potential to overcome some of the inherent limitations of such a monitoring program, e.g., with regard to temporal, vertical and horizontal coverage. In the following section 3.3, we will discuss, by means of example, some of the resulting implications for the management of water-related risks in the state of Brandenburg.

### 3.3 Implications for the management of water-related risks in Brandenburg

For the federal state of Brandenburg, the presented CRNS-based soil moisture monitoring network is the first effort to obtain soil moisture time series across important land cover types, soils, groundwater and climate conditions, at a high level of horizontal and vertical representativeness. With regard to the management of water-related risks, however, the instrumental monitoring approach itself is inherently limited. In the following, we specify these limitations, and discuss perspectives of how to address them, based on the results presented in section 3.2.

– **The observed time series are relatively short.** Although continuously growing, our observational records only start around spring 2024. For drought risk management or decision support, however, it is typically required to put the soil moisture level at a specific point in time in context with the statistical properties over longer historical periods (typically several decades). If, for instance, such a "temporal upscaling" is required *at the monitoring locations*, the bias-corrected (or calibrated) simulation models (such as ERA5-Land and, in particular, the local SWAP model) are clearly preferable (based on the evaluation of selected products in section 3.2). While ERA5-Land goes back until 1951, the SWAP model can be forced with DWD's climate station records that go back for decades, some even for more than a hundred years, or with DWD's interpolated product HYRAS-DE that reaches back to 1951. As for satellite-based products, any high-resolution products (1 km) that build on the Sentinel-1 C-SAR platform (such as the SWI) will be limited to a start year of 2015, and are hence not yet suitable to obtain any long-term statistics. At a lower resolution of 12.5 km, the SWI is available since 2007 while the low resolution C3C products reach back to 1978.

– **The monitoring network is sparse.** With only eight monitoring locations (or 12, as of November 2025), we cannot cover all relevant combinations of environmental characteristics (climate, vegetation, soil, groundwater depth), not to mention a full coverage of the state. In order to support risk assessment and management, however, the need for spatial upscaling, i.e. the prediction of soil moisture at unsampled locations, is evident. At this point, we would like to reiterate that the local bias correction is not readily transferable in space. Out of the limited number of evaluated products *without* bias correction or local calibration (Fig. 3a-d), the uncalibrated SWAP model clearly shows the highest potential (NSE of 0.68 across all locations). Homogeneous model input data such as soil texture, land use, depth to groundwater, and hydro-meteorological forcing are available for the entire state of Brandenburg. As a first upscaling application, Francke and Heistermann (2025) already used the model to assess the impact of climate change on groundwater recharge for five catchments across the state of Brandenburg. We should note, however, that the SWAP model is just one representative of physically-based hydrological models. In our opinion, the quality of the input data and the scale (horizontal and

vertical resolution) of the model application are probably more important than the model itself. Furthermore, spatial upscaling might of course benefit from the combination of different data sources, i.e. from remote sensing and modelling, probably aided by machine learning. Certainly, the spatial prediction problem remains the main challenge ahead, and the CRNS-based monitoring data will be valuable for training and validating such efforts for the state of Brandenburg. And, undoubtedly, any such efforts will benefit from additional monitoring locations that would extend the diversity of site characteristics currently covered by the network.

– **The penetration depth is limited and inhomogeneous.** $\theta_{\mathrm{CRNS}}$ provides a depth-integrated soil moisture estimate. While the penetration depth of around 30 cm is generally considered an asset of the CRNS technology, many plants (not only forests) draw their water from larger depths, so that the CRNS-based soil moisture estimates cannot generally be considered to represent "the root zone". Furthermore, the measurement depth itself depends on soil moisture and is hence dynamic (see Fig. 4). Yet, many applications in drought risk management (e.g., irrigation scheduling or drought hazard assessment) require the quantification of soil water storage down to a specific and time invariant depth that depends, for instance, on the rooting depth of the vegetation or crop of interest. Again, simulation models appear preferable to accommodate such requirements, simply because the integration of soil water storage can be handled flexibly across depths (depending on model setup). Surface soil moisture products such as C3C lack such ability, and while the exponential filtering behind the SWI aims to provide information across different depths, the results of the performance evaluation in section 3.2 speak in favour of the bias-corrected simulation models. Certainly, this implicitly assumes that the superior model performance within the CRNS penetration depth extrapolates also beyond this depth which is an admittedly strong assumption, the validity of which would have to be investigated in future studies.

– **The local water balance remains unmonitored.** For water resources management, the surface water balance is crucial for the assessment of water availability. In Brandenburg with its rather flat terrain and permeable soils, surface or near-surface runoff is rather insignificant (Francke and Heistermann, 2025), so that the surface water balance is essentially about the partitioning of precipitation between evapotranspiration and deep percolation (or groundwater recharge). In Brandenburg, this groundwater recharge is hence the key water resource that feeds surface water bodies (by means of exfiltration) and freshwater water supply (for households, industry and agriculture). Evidently, soil moisture monitoring does not directly inform us about the underlying vertical fluxes. Again, though, physically-based simulation models also represent the corresponding vertical water fluxes, and a model that performs well in capturing the observed soil water dynamics in the root zone increases our confidence in its ability to represent the surface water balance. According to section 3.2, this would again be the SWAP model. Since the calibration of the SWAP model implies a mere fine-tuning of the sand content (within the bounds defined by the soil map), we can assume that the calibrated model version is still able to consistently represent soil moisture *and* vertical fluxes. Still, an independent validation of vertical fluxes, e.g., based on eddy flux observations, would be preferable and should be a subject of prospective research.

Given these limitations, the instrumental monitoring is expected to unfold its actual value when being used to improve and assess soil moisture products and simulation models with regard to their regional applicability. The requirements to any

such product will, however, very much depend on the specific application context. For instance, irrigation management will require volumetric soil moisture estimates rather than relative saturation values, and a spatial resolution even higher than 1 km, allowing to support decisions at the plot level. In turn, irrigation scheduling might not require the availability of long time series while these are vital for drought hazard and risk assessment as well as climate impact research. When it comes to water resources management, soil moisture itself is not a target variable, but can still be valuable to improve and validate the ability of hydrological models to represent the surface water balance.

While it is beyond the scope of this study to comprehensively discuss all relevant application fields, we would like to present two examples that merely illustrate the application of the calibrated SWAP model to overcome some of the aforementioned limitations.

In our **first example**, we quantified the volumetric soil water storage for different integration depths (0-50 cm, 0-100 cm, and 0-150 cm) in the period from 1993 to 2024 (addressing the issues of limited times series length and penetration depth). Fig. 5 contrasts the development of soil water storage for selected years (the very wet year 2017, the very dry year 2022, and the first year of network operation 2024) with the seasonal dynamics of the interquartile-range (IQR, i.e. the range between the 25th and 75th percentile for each day of the year between 1993 and 2024). While the figure is very rich in details, our main point here is to demonstrate the variability of soil water storage in space (between locations) and time (between seasons and years). The highest contrast in storage is between the Kienhorst location (pine forest on middle sand with a relatively deep groundwater table) and the locations Golm and Paulinenaue (grassland on very fine sand with a shallow groundwater table). There is also a strong variability across Brandenburg in the development of soil water storage in the very wet year 2017 and the very dry year 2022: for the Oehna location, both years were close to or within the IQR while for the locations Dedelow and Marquardt, the contrast between 2017 and 2022 is very pronounced. This demonstrates the need to account for both spatial and temporal variability. The example of the year 2022 also shows how the persistence of water deficits depends on the integration depth: e.g., at the Dedelow location, storage within the upper 50 cm already approached the IQR at the end of the year 2022 while the storage in the upper 100 and 150 cm was still clearly within the lowest quartile. Conversely, at the Lindenberg location, the 2022 drought already ended in August after a series of heavy rainfall events.

In our **second example**, we keep the analysis period (1993-2024) and highlight the same selected years (wet 2017, dry 2022, and 2024); however, we look at groundwater recharge (GWR) instead of soil water storage. As pointed out above, GWR plays a crucial role for water resources management in Brandenburg. As a common proxy for GWR, Fig. 6 shows the cumulative net water flux across a soil depth of 2 m. As with soil water storage, we observe a strong variability of GWR across time (years) and space (locations). For most locations, 2024 features an exceptional level of GWR which is, to a large extent, the consequence of an extraordinarily wet end of December 2023. The peculiarity of the year 2024 becomes specifically obvious in the Kienhorst location, a dry pine forest for which annual GWR is typically close to zero. In locations with a shallow groundwater tables (most notably in Golm and Paulinenaue, less pronounced in Lindenberg), the seasonal dynamics of GWR are very different from those locations with a deep groundwater table. This is caused by an upward flux from the groundwater surface to the root zone during the summer, even causing a negative cumulative water balance in some years (most clearly in 2022). Brandenburg features extensive lowlands with shallow groundwater, so that this process is of fundamental importance for the surface water

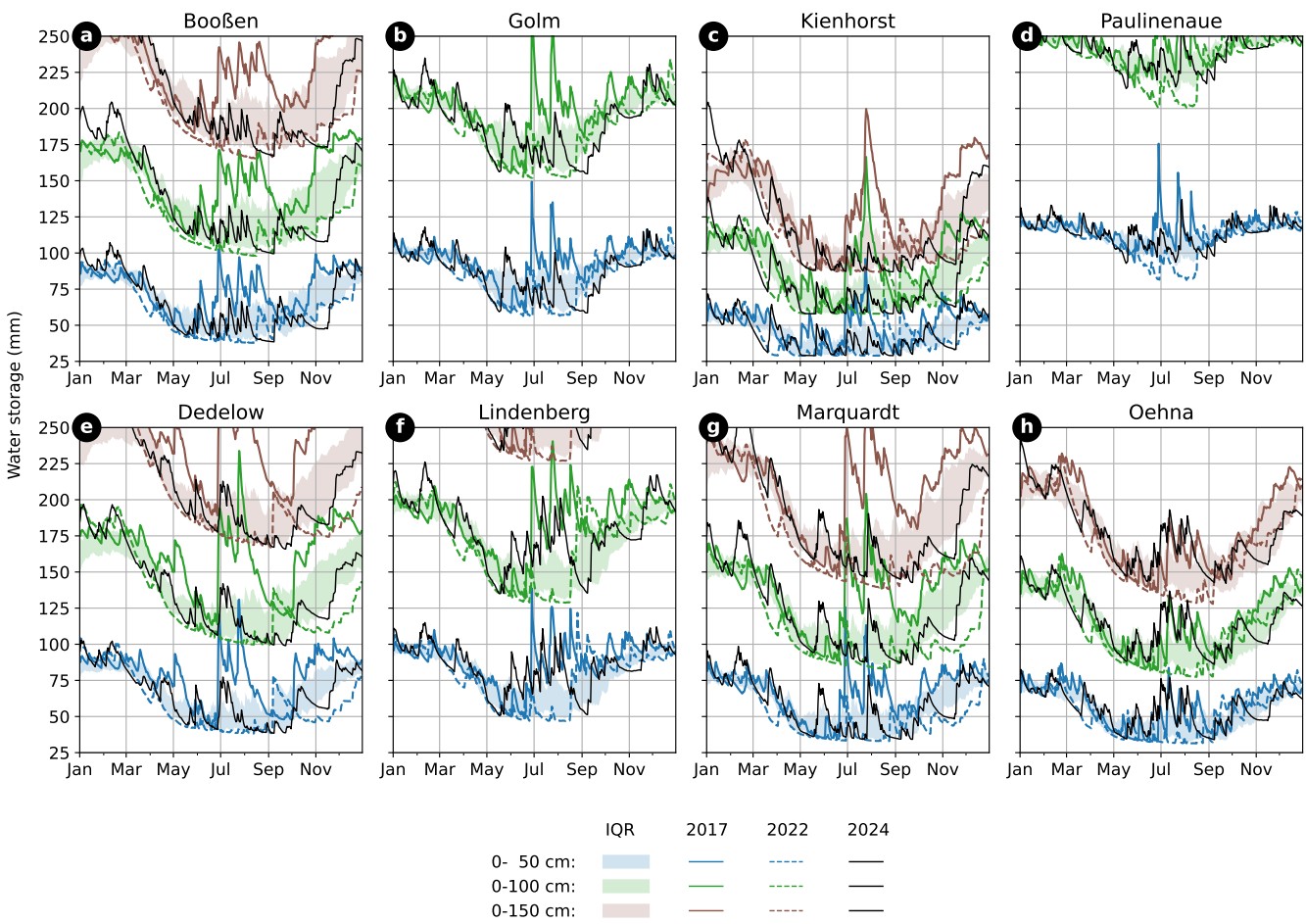

**Figure 5.** Model-derived soil water storage (in mm) for different integration depths (0-50 cm, 0-100 cm, 0-150 cm). The coloured shaded areas show the interquartile range of soil water storage for the 30 year period 1994-2023. The solid lines show the seasonal dynamics for the year 2017 (very wet), the dashed lines for the year 2022 (very dry), the black solid line for year 2024. Note that storage for the upper 150 cm is not fully shown for some locations (Golm, Paulinenaue, Lindenberg) because we used a uniform y-axis scaling (to allow comparability) while limiting the y-axis range to allow for better distinguishing temporal dynamics at different depths.

balance. It is, however, difficult to represent in large scale models: even within one kilometre, the groundwater table depth can vary dramatically. To address this issue, a hydrotope approach is hence more suitable than running a model on a grid (Francke and Heistermann, 2025). Finally, the figure highlights the remarkable development of the wet summer of 2017 in which parts of Berlin and Brandenburg were also affected by an extreme rainfall event (Caldas-Alvarez et al., 2022). That summer featured a positive net flux across the 2 m depth for the locations Golm, Paulinenaue, Dedelow, Lindenberg, and Marquardt - a process that is typically limited to the winter season. This also illustrates that, due to the low storage capacity of soils in Brandenburg,

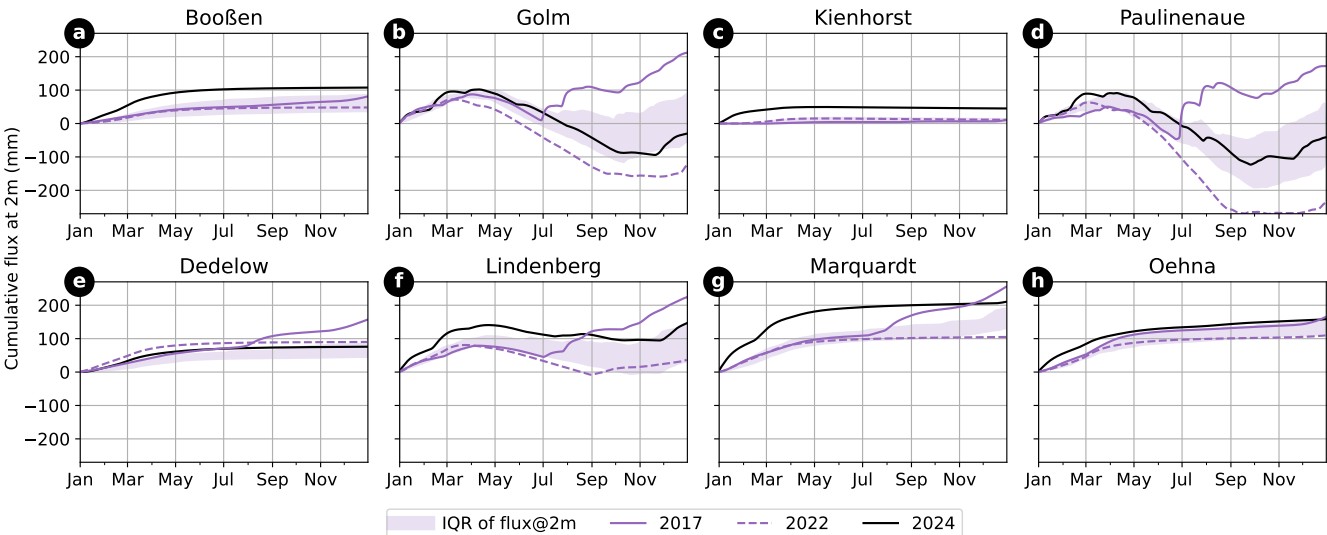

**Figure 6.** Modelled cumulative net flux across a soil depth of 2 m, as a proxy for groundwater recharge. Positive values indicate a net flux towards the groundwater table while negative value indicate a flux in the opposite direction, i.e. towards the soil surface. The purple shades shows the interquartile range of the annual cumulative flux in the period from 1994 to 2023. The lines show selected years (2017: wet year, 2022: dry year, 2024: first monitoring year).

precipitation anomalies tend to affect vertical fluxes stronger than soil water storage, and that temporal dynamics of soil water storage do not allow for any direct inference of vertical fluxes.

## 3.4 Lessons learned from the first year of operation

Apart from the aforementioned theoretical findings (sections 3.1-3.3), the first year of network operation also brought some practical and organisational experiences which we would like to share in brief.

- Working together with state government agencies from the very beginning helped to align the outcome of the monitoring effort with the requirements of the actual users, starting from the selection of monitoring locations (see also section 2.2) and not ending with the development and presentation of monitoring products. This co-design approach should also
help to make the effort more sustainable, anchoring it in institutional structures that are more long-lived than research contexts, and also taking advantage of synergies with existing monitoring infrastructures.

- Collocating the CRNS sensors with a neutron detector of known sensitivity *before* the sensors are installed in the field helps to detect, track and understand any later changes in sensitivity (e.g. from drift or firmware updates, see next points).

- As with any sensor operated under outdoor conditions, CRNS instruments are prone to a range of issues, such as, e.g.,
failures of remote data transmission in areas with poor network coverage, failures or limitations in solar power supply for

specific environments (namely forests) or seasons (namely winter), or sensor drift and instability. For the timely detection of any of these issues, it was vital to set up, from the beginning, a routine near-real time data retrieval and processing workflow, including a visualisation that allows for an intuitive detection of gaps or inhomogeneities. Specifically for CRNS sensors, this includes an early implementation of soil moisture retrieval since implausible records become more obvious for a rather intuitive variable such as soil moisture in comparison to a more complex variable such as neutron intensity.

- In the same vein, it is helpful to implement and operate a soil hydrological model for the monitoring locations as early as possible. This does not only provide an added value from the scientific perspective (as outlined in section 3.2), but also allows for the detection of more subtle sensor issues. For instance, in the context of a series of firmware updates for some of the sensors, the comparison to the routine model output allowed for a timely detection of changes in sensor sensitivity which propagated to the soil moisture estimates in a substantial, but less obvious way.

- Given the previous two items, we set up a platform for visualising and sharing both observational and simulated data (https://cosmic-sense.github.io/brandenburg) with relatively short latency. The platform is under continuous development, specifically with regard to data presentation formats, and open to suggestions by interested users.

## 4   Conclusions and outlook

In this study, we introduced a new network for long-term soil moisture monitoring in Brandenburg, using cosmic-ray neutron sensing (CRNS) technology. The launch of this network resulted from a joint effort of research institutions and state government agencies. In 2024, eight locations were instrumented, and four more will follow in November 2025.

We consider the monitoring network as an important asset to support the management of water-related risks in Brandenburg, as it represents the typical regional characteristics in terms of climate, soils, land use, and distance to the groundwater table. Beyond the analysis of the monitoring data alone, the observational records can be leveraged to develop and evaluate soil moisture products that could overcome the limited temporal, vertical and horizontal coverage of the network. As a corresponding case study, we assessed selected large-scale soil moisture products from modelling (ERA5-Land), remote sensing (C3C) and combinations of both (SWI) as well as a local soil hydrological model (SWAP) with regard to their ability to capture the observed local soil water dynamics in Brandenburg. The pure modelling products, namely ERA5-Land and, in particular, SWAP, clearly outperformed the satellite-based products for both cases, with and without local bias correction or calibration. While the uncalibrated SWAP model is most promising for the regionalisation of soil moisture, the calibrated version appears most suitable for the long-term analysis of soil water storage and the surface water balance at the monitoring locations.

Rather than as a final analysis, this work should be seen as a starting point to demonstrate the potential of the CRNS-based soil moisture estimates. In order to stimulate future applications in various related fields, and to allow for any interested parties to use the data according to their priorities, we openly share the observational and the simulated data on a public platform (see

section "Data availability"), and invite collaboration in the improvement, enhancement, and integration of our network. That way, various opportunities arise, which could include, but are not limited to:

- **Improving soil moisture retrieval from CRNS**: in close collaboration with the sensor manufacturers, the long-term operation of CRNS sensors should help us to identify, understand and fix sensor issues that are, e.g., related to signal stability and traceability. Furthermore, there is a considerable potential to further improve the CRNS-based soil moisture estimation. For the relatively dry conditions in Brandenburg, a new conversion function recently suggested by Köhli et al. (2021) appears particularly promising. To that end, it would be desirable to combine this function with attempts to generalize the estimation of soil moisture from neutron intensities (Heistermann et al., 2024).

- **User-oriented monitoring products**: The integration of model and observation should allow to custom-tailor data products to specific user requirements. For instance, different vertical integration depths or temporal aggregation levels of soil water storage might be relevant in the context of wild fire hazards, agricultural management (e.g., timing of field operations, including irrigation), water resources management, or flood hazards (e.g., capacity for soil water retention, although flood generation is not a primary concern in Brandenburg). The design of such products should be subject to a continuous dialogue with potential users in the aforementioned sectors, including the involved federal state agencies, but also, e.g., farming or forestry companies.

- **Groundwater recharge**: Similarly, combining model and observational data should enable more accurate estimates of groundwater recharge rates under different conditions, including the analysis of land use and climate change. This is a key challenge for water resources management in the state of Brandenburg (Francke and Heistermann, 2025; Somogyvári et al., 2025).

- **Spatial upscaling**: in our view, spatial upscaling or regionalisation remains the main challenge in soil moisture monitoring. To that end, the soil hydrological model SWAP, set up with region-specific data, clearly showed the best prospects, at least for locations that are (in terms of climate, soil, land use and groundwater table depth) similar to the locations represented by the monitoring network. Rather than overemphasising the success of this model in our admittedly limited benchmarking case study, we would like to reiterate that the SWAP model is just one representative of physically-based hydrological models, and that, in our opinion, the quality of the input data and the scale (horizontal and vertical resolution) of the model application are probably more important than the model itself. Furthermore, combinations of remote-sensing data and simulation models, e.g., by means of assimilation or machine learning, might unlock further predictive skill of satellite products that has remained hidden at least in our analysis. Here, an emerging perspective for future research is rail-borne CRNS roving: several locomotives of a regional rail company in Brandenburg have recently been equipped with CRNS sensors in order to monitor spatio-temporal soil moisture patterns along selected railway tracks (see Fig. 1). While those of our monitoring locations which are close to these railway tracks (Tab. 1) could be used to verify the spatiotemporal integrity of the railborne data products, the latter could, in turn, be used to train or validate other model-based or data-driven upscaling approaches.

– **Network extension**: it is planned to expand the network in Brandenburg, and we encourage other institutions to integrate their sensors or to propose suitable locations for deployment. Similar efforts are underway in other federal states, e.g., Saxony (Schrön et al., 2025). Collaboration and integration with these initiatives could be a pathway towards a prospective nation-wide monitoring.

The CRNS-based soil moisture monitoring network is intended as a long-term activity, and will also increase its value as the
length of the observational time series increases and hence covers a higher diversity of hydro-climatological conditions.

*Data availability.* The observed neutron time series as well as the soil moisture products (as retrieved from neutron counts and modelling) are openly availably at https://doi.org/10.23728/b2share.dfde74f4be294bd7b927f67988365f8e (Heistermann et al., 2025). Furthermore, raw CRNS observations, CRNS-based soil moisture estimates, and simulated soil water content are openly and continuously available for download at https://cosmic-sense.github.io/brandenburg.

*Author contributions.* MH, DA, TF, MS, SA, PD and SO were involved in the conceptualization of the study, DA, PMG, AB, AM, and SO designed and established the CRNS monitoring network, with contributions from MS, MH, TF, and PB. TF designed the soil sampling campaigns. MH, TF, PB and PMG were responsible for data curation, with support by JT. SO, SA, SZ, PD, DA, MH, and TF contributed to funding acquisition and project administration. RE, FB, and AW supervised and mentored the project. MH carried out the data analysis and modelling, created the figures and drafted the manuscript, with support from TF. All co-authors contributed to writing, reviewing and editing
the manuscript.

*Competing interests.* The contact author has declared that neither they nor their co-authors have any competing interests.

*Acknowledgements.* This research was funded by the Helmholtz-Centre for Environmental Research GmbH - UFZ, the Ministerium für Landwirtschaft, Umwelt und Klimaschutz des Landes Brandenburg (MLUK, Ministry of Agriculture, Environment, and Climate Protection of the federal state of Brandenburg) as well as by the Deutsche Forschungsgemeinschaft (DFG, German Research Foundation) – research
unit FOR 2694 "Cosmic Sense", project number 357874777.

We gratefully acknowledge the landowners and organisations granting access to the sites: Axel Behrendt, Gernot Verch (ZALF), Rainer Hentschel, Paul Reibetanz, Jens Hanneman (State Forestry Office Brandenburg), Ronny Henkel (Rail & Logistik Center Wustermark), Agro Uetz-Bornim GmbH, Oehnaland Agrar GmbH, Christiane Sengebusch, and Philip Golo. The authors furthermore acknowledge the support from the Havelländische Eisenbahn Gesellschaft (HVLE), Wustermark, Germany, particularly from Dirk Brandenburg, Uwe Wullstein,
Bastian Weber, Oliver Georgius and Stadler Rail, David Sorribes.

We thank Markus Köhli and Jannnis Weimar (StyX Neutronica) for technical support with some of the sensors.

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
