# Peer review of "Soil moisture monitoring with cosmogenic neutrons: an asset for the development and assessment of soil moisture products in the state of Brandenburg (Germany)"

_EGUsphere, 2024_

## Author Comment (AC1)

**Interactive Discussion: Author Response to Referee #1**

**Brief Communication: A new drought monitoring network in the state of Brandenburg (Germany) using cosmic-ray neutron sensing**

D. Altdorff, M. Heistermann, T. Francke, M. Schrön, S. Attinger, A. Bauriegel, F. Beyrich, P. Biró, P. Dietrich, R. Eichstädt, P. M. Grosse, A. Markert, J. Terschlüsen, A. Walz, S. Zacharias, S. E. Oswald

*NHESS Discussions,* `doi:10.5194/egusphere-2024-3848`
* * *
**RC:** *Reviewer Comment*,     AR: *Author Response*,     ☐ Manuscript text

Dear madam or sir,

thank you very much for your referee report, and for the time and effort you spent to examine the manuscript.

We understand your concerns, will try to address them in a revised version of the manuscript. At the same time, we hope that this response can contribute to a better understanding of the context and scope of the Brief Communication manuscript. Please find both your comments and our responses below in a point-by-point reply.

Thanks again for your feedback and your support of this process.

Kind regards,
Maik Heistermann (on behalf of the author team)

**RC:** *The Authors present a very relevant and interesting initiative to establish an operational ground-based long-term soil moisture monitoring based on cosmic-ray neutron sensors (CRNS). Nine locations and around six months of data are presented. Comparisons with the soil moisture simulated by the agro-hydrological model SWAP are also reported. The manuscript reads well and clear. The preliminary results are meaningful and the discussion fair. Personally, I believe that the main contribution of this initiative is (L39) to bring together a consortium of research institutions and state agencies for establishing a long-term monitoring. And I would congratulate with the Authors for such an effort in moving CRNS research activities towards long term operational monitoring.*

**AR:** Thank you so far...

**RC:** *But back to the manuscript, it reads like an internal report of current status of the initiative and I'm not convinced that is worth a publication. I admit that this bold statement starts from the wish to read novelty in scientific papers and to not consider pure tech-transfer project innovative. In questioning my self-position, I made two actions. First, I look at the description of Brief communications, as it could have provided me with the right angle for judging the manuscript. The description is reported below. In addition, I looked at other published papers with similar vision (to my knowledge). Some of the papers are listed below (Benninga et al., 2018; Cosh et al., 2021). As for the Brief Communications, I leave to the Handling Editor to decide. Anyway, I highlight how the number of pages of the manuscript is way too much. Moreover, its scope might be (c) even if information and data do not seem to be properly disseminated.*

Brief communications are timely, peer-reviewed, and short (2–4 journal pages). These may be used to (a) report new developments, significant advances, and novel aspects of experimental and theoretical methods and techniques which are relevant for scientific investigations within the journal scope; (b) report/discuss significant matters of policy and perspective related to the science of the journal, including "personal" commentary; (c) disseminate information and data on topical events of significant scientific and/or social interest within the scope of the journal. Brief communications have a maximum of three figures and/or tables, maximum 20 references, and an abstract length not exceeding 100 words. The manuscript title must start with "Brief communication:".

AR:  We thank the referee for the critical appraisal of the manuscript. While we appreciate the positive feedback regarding relevance, readability, clarity, and meaningfulness, we also understand the concerns. Trying to rephrase these concerns in our own words, the referee essentially argues that, for a brief communication, the paper is too long and lacks a proper match to the typical use case of this article format; for a research article, in turn, the paper lacks novelty and depth.

We also appreciate the referee's efforts to reflect the requirements to a Brief Communication (as provided by NHESS), and to extend the view to other articles that introduced soil moisture monitoring networks (Benninga et al., 2018; Cosh et al., 2021).

However, we would like to use this opportunity to put our manuscript and the related efforts into context, and to justify the chosen publication forum and format – as it was a deliberate and well-founded decision to use the Brief Communication format the way we did.

To understand this decision, though, an important aspect is missing in the assessment of our manuscript. This missing aspect is the context of the special issue to which our manuscript was submitted, entitled "Current and future water-related risks in the Berlin–Brandenburg region" (see `https://nhess.copernicus.org/articles/special_issue1295.html`).

In our manuscript, we clearly elaborated how our soil moisture monitoring network relates to the topics of the special issue, i.e. by providing support to "making human-environment systems more resilient to water-related hazards", to create "forms of governance to cope with emerging challenges", and to raise "public awareness [...] and appropriate adaptation practices" (quotes are excerpts from the special issue's topics). As important as this topical match, however, is the explicit and distinct regional focus of the special issue on the federal state of Brandenburg – which is, as one of the driest states in Germany, increasingly challenged by issues of water scarcity.

We are convinced that a new initiative on soil moisture monitoring *in Brandenburg* which is rooted in both research and public administration is worth being reported in this very special issue of NHESS. We even think that it would be negligent *not* to report on this initiative in such a context.

At the same time, we couldn't agree more with the referee that putting together a monitoring network – time consuming and expensive as it may be – is *not* a scientific achievement in itself and does hence not qualify, in a scientific sense, as "novel" or "innovative". Then again, our initiative is certainly novel for the state of Brandenburg in the sense that it opens new opportunities for research and applications towards the aims outlined in the special issue (see above).

This context is exactly why we preferred the format of a Brief Communication over the format of a research article: our manuscript is *"timely [...] and short"*, and it aims to *"report new developments [...] relevant for scientific investigations within the journal scope"* and to *"disseminate information and data on topical*

*events of significant [...] interest within the scope of the journal"*. Here, the mentioned scope is specifically the scope of the special issue. In other words: We chose the Brief Communication format *intentionally* and exactly because we do not want to claim scientific novelty. Instead, our intention is to inform researchers and practitioners in various fields about a new perspective to improve the management of water-related risks in Brandenburg. The special issue is a unique opportunity to inform this specific audience, and we believe that we will reach it more efficiently with a *brief* communication without excessively reporting on scientific and technical details. We also believe that this is a good timing, as it gives interested institutions and people the opportunity to engage rather sooner than later.

We hope that our (admittedly verbose) explanation provides the required context to assess the relevance of our manuscript and the adequacy of our choice of the article format. On that basis, we would like to address some of the more specific issues raised by the referee:

**RC:** *I highlight how the number of pages of the manuscript is way too much.*

 AR: We would like to clarify that the length requirement for Brief Communications applies to the final, typeset version, which is significantly shorter than the preprint in terms of number of pages. Our preprint currently spans 10.5 pages, which is even slightly below the average length of preprints for Brief Communications that were accepted for publication in NHESS over the past two years.

We hope this explanation addresses the referee's concern, and demonstrates how our manuscript aligns with the journal's requirements. Still, we will try, in a potential revision, to shorten the manuscript further, without compromising the clarity and completeness of our argument. Specifically, we could shorten the description of the criteria for site selection (ll. 54-71) and the presentation of the case study (section 4).

**RC:** *Moreover, its scope might be (c) even if information and data do not seem to be properly disseminated.*

 AR: As elaborated in our above response, we think that the scope of our paper relates to items (a) and (c) in the description of Brief Communications. Furthermore, we emphasized above why we think that the dissemination of information exactly serves the desired purpose: to provide a brief, timely and comprehensible synopsis of our initiative in order to invite interested institutions and users *now*, and to use the channel of the special issue to reach exactly the audience that is involved with the research and applications related to the management of water-related risks in Brandenburg.

**RC:** *By looking at other published papers on similar topic (examples above), I have seen much more information in describing the issues and the effort, e.g., to establish such a network, to standardize the observations, to integrate the new data in current platforms, in defining accessibility to end users.*

 AR: We outlined the motivation of our paper above, and would like to reiterate that it is *not* the aim of our manuscript to systematically share experiences with the establishment and maintenance of a sensor network, to push forward efforts for standardizing data and data processing, or to define novel channels and formats for data dissemination. And while we appreciate the reference to Benninga et al. (2018) and Cosh et al. (2021), we would like to maintain that these are different types of papers with different motivations. Benninga et al. (2018) is a data paper, the purpose of which is to comprehensively document a defined dataset. The dataset reported in Benninga et al. (2018) does not refer to a continuous monitoring effort, but to a fixed period (April 2016 to April 2017), and was, afterwards, only updated once in 2018 and discontinued afterwards. Cosh et al. (2021), in turn, reported on a massive strategic effort at the national scale of the US which aims at integrating all kinds of data, with substantial infrastructural support. We do not consider it helpful to compare that kind of paper to our comparatively light-weight approach that is tailored to the requirements of the federal state of Brandenburg. Having said all that, we will, in our response to the next comment, elaborate how we could include some of our experiences in a concise way in the present manuscript format.

**RC:** *Overall, while I'm currently not in favour of the publication of the present manuscript, by considering previous published papers, it might be considered if the Authors put much more effort and they succeed in improving the manuscript, e.g., by sharing their experiences in establishing, maintaining, and managing a fixed environmental sensor network that could be of utility to the community for avoiding mistakes and reproducing good practices.*

**AR:** We appreciate the suggestions for improving the manuscript, and we agree that the referee raises several issues that are of interest specifically on a technical level for similar initiatives aiming to establish such networks. However, we are not convinced that including such aspects would improve the present manuscript, for the following reasons: we have shown above that our manuscript meets the requirements of a Brief Communication. However, by adding the requested details, the paper will not be a Brief Communication anymore. But for serving the desired purpose – which we comprehensively elaborated above –, the Brief Communication format is ideal and should hence not be inflated with excessive detail.

But while it appears difficult to resolve the trade-off between length and depth in the framework of a Brief Communication, we agree that it could be helpful and manageable to outline some important "lessons learned" in the bullet point "network extension" after l. 151 of the preprint – since this paragraph in fact aims to build a bridge to similar ongoing efforts in neighboring federal states in Germany. We will slightly extend the paragraph accordingly in a revised version.

**RC:** *Improving metadata (e.g., how was the calibration performed?), transparency (how data have been processed) and data accessibility (e.g., by API-type) should also justify the publication.*

**AR:** As pointed out before, we think that it will not serve the manuscript well to add much more technical detail with regard to data processing and analysis. However, we agree it would be helpful to clarify some details with regard to the estimation of soil water content from the neutron counts. While the preprint has already referred to a publication (Heistermann et al., 2024) which outlines the applied concept of a general CRNS calibration (in order to avoid uncertainties from the strategy of a local calibration), we agree that, in a revised version, this could be stated more clearly. In order to enhance transparency, we will also publish the code repository containing the data processing scripts upon final publication of the manuscript. It should, however, be very clear that our estimates of soil water content are only one possible way to obtain a product, and that other users might prefer to apply other procedures.

For this reason we established, at the very beginning, a public directory to download both, raw observations *and* our soil moisture product. The link to that directory is provided on the website that we already referred to in the preprint, and it allows to access and view that directory via a browser interface. It also allows to obtain static https-links to each of the data files which can be directly used in any automatic data processing environment and correspond, in our view, to what the referee refers to as "API-type" access. The directory also contains station metadata in a dedicated table.

The open accessibility of data and metadata under that link was clearly stated in the preprint under the section "data availability". Please note, however, that we are currently revising the data management after a major DoS attack on the entire university end of 2024, so that the updating of the website and the data directory have been put on a hold for a few weeks (as also announced on the website).

**References**

Benninga, H.-J. F., C. D. U. Carranza, M. Pezij, P. van Santen, M. J. van der Ploeg, D. C. M. Augustijn, and

R. van der Velde (2018): The Raam Regional Soil Moisture Monitoring Network in the Netherlands. Earth System Science Data 10, 1, 61-79. https://doi.org/10.5194/essd-10-61-2018.

Cosh, M. H., T. G. Caldwell, C. B. Baker, J. D. Bolten, N. Edwards, P. Goble, H. Hofman, et al. (2021): Developing a Strategy for the National Coordinated Soil Moisture Monitoring Network. Vadose Zone Journal 20, 4, e20139. https://doi.org/10.1002/vzj2.20139.

Heistermann, M., Francke, T., Schrön, M., and Oswald, S. E. (2024): Technical Note: Revisiting the general calibration of cosmic-ray neutron sensors to estimate soil water content, Hydrol. Earth Syst. Sci., 28, 989–1000, https://doi.org/10.5194/hess-28-989-2024.

---

## Author Comment (AC2)

**Interactive Discussion: Author Response to Referee #2**

**Brief Communication: A new drought monitoring network in the state of Brandenburg (Germany) using cosmic-ray neutron sensing**

D. Altdorff, M. Heistermann, T. Francke, M. Schrön, S. Attinger, A. Bauriegel, F. Beyrich, P. Biró, P. Dietrich, R. Eichstädt, P. M. Grosse, A. Markert, J. Terschlüsen, A. Walz, S. Zacharias, S. E. Oswald

*NHESS Discussions,* `doi:10.5194/egusphere-2024-3848`
* * *
**RC:** *Reviewer Comment*,    AR: *Author Response*,    ☐ Manuscript text

Dear madam or sir,

thank you very much for your referee report, and for the time and effort you spent to examine the manuscript.

We understand your concerns, and will try to address them in a revised version of the manuscript. In the same way as in our response to referee #1, though, we also hope that this response can contribute to a better understanding of the context and scope of our manuscript. Please find both your comments and our responses below in a point-by-point reply.

Thanks again for your feedback and your support of this process.

Kind regards,
Maik Heistermann (on behalf of the author team)

**RC:** *In this Brief Communication, the authors present a new initiative in Germany to establish an operational, long-term soil moisture monitoring system based on cosmic-ray neutron sensors (CRNS) for the state of Brandenburg. In general, this is a very positive development, as CRNS has established itself as a new standard method for the continuous determination of soil moisture at the field scale and the widespread deployment of these sensors is very helpful and highly welcome both for environmental research and for the support of water management. In terms of content, this article presents the status of the project as well as a comparison of the measured soil moisture with simulations of the agrohydrological model SWAP. The manuscript is well written [...]*

**AR:** We thank the referee for the positive feedback on the relevance of our initiative and the readability of the manuscript which are, in our view, important aspects of the evaluation.

**RC:** *[...] however, I have to agree with reviewer #1 that the paper reads more like a project report and does not fulfill the NHESS requirements for short communications, especially in terms of the length of the paper.*

**AR:** We also appreciate the critical appraisal. Since the referee explicitly refers to the arguments outlined in the report of referee #1, we would also like to explicitly refer to our response to referee #1 (link), and keep the response in this context a little bit shorter. Still, we would like to specify (and partly repeat) some aspects here, too:

- Regarding the statement that the paper "does not fulfill the NHESS requirements for short communications, especially in terms of the length of the paper", we would like to maintain that the length requirement for Brief Communications applies to the final, typeset version, which is significantly shorter than the preprint in terms of number of pages. With 10.5 pages, our preprint is even slightly below the average length of preprints for Brief Communications that were accepted for publication in NHESS over the past two years.

- Still, we are really willing to scrutinize, without reservation, the manuscript in order to find and remove details that might not be essential to our overall argument. The referee's specific comments (below) contain some helpful guidance as to where the text could be shortened (e.g. around ll. 54-71 of the preprint). As already pointed out to referee 1, we also see potential to shorten the section about the case study.

- We are not sure how to interpret the statement that the paper "reads more like a project report". Is this meant in the sense that project reports tend to be less concise – to report details that might not be considered as scientifically relevant? This interpretation would be consistent with the referee's assessment that the paper is too long for a Brief Communication (which we addressed in our above response above). We kindly ask for clarification if this comment was meant in a different way.

**RC:** *Also, it may be beneficial to complete the installation and subsequently measure soil moisture over a longer period of time. Once this has been achieved, the authors could publish the results as a data paper.*

AR: As mentioned before, the motivation of the brief communication is to raise awareness for this initiative among researchers and experts in the field of management of water-related risks in Brandenburg, and to use the opportunity to reach this audience, as presented by the very specific topical and regional scope of the special issue to which we submitted our manuscript (entitled "Current and future water-related risks in the Berlin–Brandenburg region", see `https://nhess.copernicus.org/articles/special_issue1295.html`). As already pointed out to referee 1, we are convinced that a new initiative on soil moisture monitoring *in Brandenburg* which is rooted in both research and public administration is worth being reported in this special issue of NHESS (both referees, in their reports, explicitly welcomed and appreciated this development). We even think that it would be negligent *not* to report on this initiative in such a context. We are further convinced that a Brief Communication is exactly the format to bring across this message, and that our manuscript, as demonstrated above, meets the requirements of the Brief Communication format.

We would like to emphasize that our paper is *not* intended to serve as a kind of replacement for a data paper, although we provide all means to access and use both the raw measurement data and the soil moisture products already at the current stage (see also response to referee 1, and the section on data availability in the preprint). We appreciate the suggestion that a longer time series could be published as a data publication in the future, but it should also be noted that data publications refer to a closed set of data with a fixed DOI which is not what we intend in terms of continuous long-term monitoring.

**RC:** *Finally, we don't learn much from the case study and the context of the comparison of CRNS data with simulation is unclear (see also specific comment further below).*

AR: We appreciate the critical appraisal of the case study. What we take from this comment is that we need to better explain – also in paper – why we think the presentation of this comparison is helpful. At the same time, the section about the case study is quite long, compared to the other sections of the manuscript. Given our aforementioned willingness to shorten this section, the challenge for a revision of the manuscript will be to better justify the case study while dropping some technical detail.

As for justification, we would already like to provide some explanations here: Hydrological models (of which SWAP is merely an example) are the most promising option to solve three issues which an instrumental monitoring of soil moisture alone cannot solve: a) the derivation of water fluxes (such as groundwater recharge, which cannot be directly derived from soil moisture time series), b) the upscaling of soil moisture and fluxes to unmonitored locations, and c) to allow for the comparison of model and observations in order to reveal (and ideally address) errors or uncertainties in either. While this challenge is at the heart of our initiative, solving it is obviously beyond the scope of our Brief Communication.

What we can provide, though, is a glimpse – a demonstration of perspective. The consistency between our model implementation and the observations obtained so far is that kind of glimpse. We find this approach much more concrete than just pointing to the theoretical perspective of bringing together data and model.

All this information is essentially already contained in section 4. Yet, we suggest the following changes in order to clarify the motivation and still shorten the section:

- to revise the first paragraph (ll. 86-88) in order to clarify the aforementioned motivation.

- to remove technical details provided along ll. 89-100.

- some content from the description/discussion of Fig. 2 (ll. 101-124) could be shortened and moved to the figure caption, some aspects might be dropped, and the remaining main text could be further condensed.

**RC:** *L38: Why mention only national networks?*

AR: In the light of the next referee comment (reg. l. 44), we agree that the CRNS-based soil moisture monitoring network in the context of the ADAPTER project should be mentioned here, too. We will add a corresponding sentence which will also refer to Ney et al. (2021).

**RC:** *L44: Such an initiative is not unique in Germany, see e.g. Ney et al. (2021).*

AR: Given our response to the previous comment, we agree that it is difficult to maintain that our effort is unique. We would, however, like to explain the reasons why we maybe not adequately acknowledged the ADAPTER project before: Our notion was that the focus of the ADAPTER project was exclusively on agricultural areas while our initiative aims to cover forest and grassland sites, too. Furthermore, we consider the close involvement of various state agencies along the entire process as a distinct feature of our initiative. However, we agree that the notion of "uniqueness" is probably unnecessarily bold, maybe even misleading. We will hence change the sentence to

> In a recent effort, five institutions have combined their resources to establish [...].

**RC:** *Figure 1: The small maps are hardly recognizable and the situation at the measuring stations is not readable. Also, this this information is already available in table 1. Therefore, I suggest removing the small maps.*

AR: We thank the referee for the suggestion. The motivation of the maps (including the smaller ones) was *not* to show the landscape attributes at the monitoring locations. This information is, as the referee correctly pointed out, provided in Tab. 1. The idea of the small maps was rather to give an impression of the overall spatial distribution of important landscape attributes in Brandenburg. We would still prefer to show this information

as it – although implicitly – relates to the important issues of regionalisation and representativeness. In order to avoid the misunderstanding that the maps should show the attributes at the monitoring locations, we suggest that, in a revised version, we simply remove the location markers from the small maps.

**RC:** *L54-71: This sounds very much like an interim report.*

AR: As already pointed out above, we find it difficult to assess what this statement is supposed to imply. Our guess is that the referee considers this part as not sufficiently concise or relevant. If the statement was meant otherwise, we kindly ask for further guidance.

While we think that an objective documentation and justification of criteria for site selection is quite relevant, we also understand that this might be the kind of information that is not crucial to bring across our main points. We hence agree to shorten this part in a revised version of the manuscript.

**RC:** *L116-119: It is not clear to me how this comparison is to be analyzed in this project. What are the consequences if there are discrepancies? Do you then not trust the measurement data, even though the model certainly may have a much higher uncertainty?*

AR: We thank the referee for this remark. It is in fact difficult, for these lines, to find a balance between brevity and depth. First of all, we want to make the point that any discrepancy between model and observation is an opportunity to learn something (this may sound corny, but it is true). While this typically means that observations can help us to learn something about model errors (and how to address them), it could also be the other way around. In our context, inconsistencies between model and observations repeatedly helped us to detect sensor malfunctioning that did not produce obviously implausible observations. While this might be one of the "lessons learned" that referee 1 could be interested in, any detailed account of such cases would unnecessarily inflate the manuscript. However, we suggest to add, in the revised version, a similar sentence at the end of the paragraph in l. 119 in order to clarify the statement:

> In fact, inconsistencies between model and observation repeatedly helped us to detect and fix sensor issues that might otherwise have remained hidden as they did not result into obviously implausible observations.

**RC:** *L126-129: Describing commitments should not be part of a scientific publication. It would make more sense to present the concrete implementation of a real-time data platform.*

AR: One of the aims of our paper is to invite institutions and people to engage in this initiative, e.g., as users of data, in the development of helpful data products, in the dissemination, in the provision of additional monitoring locations, or by informing us of their specific needs (e.g., with regard to locations, data, or presentation of results). That is why we consider it very important to emphasize these commitments, even if this may not be the sound of a research article – because this is not a research article (at the risk of excessive repetition: we deliberately preferred the format of a Brief Communication over a Research Article).

As for the presentation of a real-time data platform, the amount and complexity of the provided data and products does not yet call for the design of real-time data platforms which would deserve a comprehensive documentation in the context of a journal article. We made a clear statement on data availability in the main text, and, more specifically, in the section on data availability. There, we provide the link to a website that includes a public directory to download both, raw observations *and* soil moisture products. You can access and view that directory via a browser interface, but you can also obtain static https-links to each of the data files which can be directly used in any automatic data processing environment. The directory also contains station metadata in a dedicated table.

As pointed out to referee 1, this website was put on a hold for a few weeks; yet, we think that it well serves the intended purpose for the time being. As the diversity, complexity and lengths of observations and products may increase, we are certainly willing to adjust this approach. However, we do not think that a detailed account of that website will serve the paper well. However, since the availability of that website and the data might not have become clear enough from the preprint, we will extend the section of data availability further, and also add to the caption of Fig. 2 the information that similar time series and figures are available on the website in near real-time.

**RC:**   *L139-140: Formulations such as "some validity" and "some transferability" are too vague.*

AR:   We agree. In a revised version of the manuscript, we will rephrase as follows:

> [...] the results of our case study demonstrate that the parameterisation of our model – as based on region-specific vegetation parameters, pedo-transfer functions in combination with regional maps of soil properties, and a map of groundwater depth – already allowed to capture soil water dynamics at the monitoring locations fairly well. This increases our confidence that the parameterisation concept could be transferred at least to similar locations, which needs to be further investigated in prospective research.

**RC:**   *L142-145: This seems to be rather an alternative measurement approach instead of an upscaling approach for the presented CRNS network.*

AR:   We agree that the description should be clarified in order to convey a better idea how the railborne CRNS roving could support the upscaling of soil moisture. We hence suggest to change, in a revised version, these lines as follows:

> An emerging perspective for future research is rail-borne CRNS roving: several locomotives of the Havelländische Eisenbahn AG have recently been equipped with CRNS sensors in order to monitor spatio-temporal soil moisture patterns along selected railway tracks (see Fig. 1). While those of our network locations which are close to these railway tracks (Tab. 1) could be used to verify the spatiotemporal integrity of the railborne data products, the latter could, in turn, be used to validate or train other model-based or data-driven upscaling approaches.

We hope that the given explanations together with the proposed outlined changes help to better put the intention of the manuscript in context, and further convince the reviewers of its merit. We are looking forward to any fruitful discussion that may yet to come.

**References**

Ney, P., Köhli, M., Bogena, H., Goergen, K. (2021). CRNS-based monitoring technologies for a weather and climate-resilient agriculture: Realization by the ADAPTER project. In 2021 IEEE International Workshop on Metrology for Agriculture and Forestry (MetroAgriFor) (pp. 203-208). IEEE.

---

## Author Response (AR1)

**Author Response Letter to Editor and Referees**

**A new drought monitoring network in the state of Brandenburg (Germany) using cosmic-ray neutron sensing**

D. Altdorff, M. Heistermann, T. Francke, M. Schrön, S. Attinger, A. Bauriegel, F. Beyrich, P. Biró, P. Dietrich, R. Eichstädt, P. M. Grosse, A. Markert, J. Terschlüsen, A. Walz, S. Zacharias, S. E. Oswald

*NHESS Discussions,* `doi:10.5194/egusphere-2024-3848`
* * *
**EC/RC:** *Editor/referee comment*,     AR: *Author Response*,     ☐ Manuscript text

Dear Dr Somogyvári, dear referees,

we would like to thank again both referees for the critical review, and the editor for the balanced decision on how to proceed with the manuscript. Following the editorial decision, we expanded our work to a research paper, and revised the manuscript accordingly.

Please find below our point-by-point replies. In our reply to the editor (section 1), we will summarize our overall revision approach, based on the outline we had already shared with the editor on February 13.

The point-by-point replies to the referees (sections 2 and 3) are partly different from our responses in the interactive discussion where we had put more emphasis on justifying the "Brief Communication" format. Instead, the responses in this letter focus more on how the actual revision of our manuscript relates to the referee comments. We are confident that we have, this way, addressed the referees' original concerns.

Thanks again for your feedback and your support of this process.

Kind regards,
Maik Heistermann
(on behalf of the author team)

**1. Response to the editor**

**EC:** *Thank you for your submission, and for answering the reviewer comments in detail. Both reviewers were very critical of multiple aspects of the submission, with the main criticism towards the Brief communication format, and whether it is the right solution for the contribution. I agree with the authors that formally the manuscript fits the BC format, and its scope also fits some of the BC criteria. However, the manuscript reads a lot like a technical note, with a lot of technical details and limited explanation, with a feeling that details had to be omitted to fit such format. Both reviewers are asking for further explanations regarding the methodology and the results - which is again pointing away from the BC format. Therefore as a BC I find it difficult to recommend the paper for further revisions. My recommendation to the authors would be instead to consider extending the manuscript to a research paper. This way, the revision could address the requested improvements from the reviewers, with special focus on the methodology, validation of the results and further details on the case study. A more detailed text would also allow for findings in the regional context, which would be relevant for the SI.*

**AR:** Based the editorial decision, we expanded our work to a research paper, and revised the manuscript accordingly. The following enumeration summarizes our overall revision approach, based on the outline we had already shared with the editor on February 13:

1. We extended the scope and the aims of the manuscript. The original (and still main) objective - to introduce the new CRNS-based soil moisture monitoring network in Brandenburg to the community, as a contribution for future research and applications - was supplemented by the following objectives: (1) to provide an evaluation of our procedure to estimate soil moisture from observed neutron counts, based on independent observations; (2) to evaluate the performance of our soil hydrological model, and on that basis, to demonstrate its potential to mitigate some of the inherent limitations of a sparse instrumental monitoring network (e.g., with regard to temporal and spatial coverage); (3) to discuss practical lessons learned from the establishment and operation of the network, as well as potential future applications.

2. On the basis of the extended scope, we added a new section 2 ("Data and methods") that combines parts of the original sections 2 and 3, and provides additional details on the the network design (section 2.1), the procedure to estimate soil moisture from neutron intensities and evaluate it by reference observations (section 2.2), as well as the set-up of the hydrological model and the metrics to evaluate its performance (section 2.3).

3. The original section 4 was comprehensively reworked into section 3 "Results and discussion" that now consists of three subsections which address the three additional objectives of our study (as outlined in item 1): the evaluation of the CRNS-based soil moisture estimation, the evaluation of the hydrological model performance, the discussion of how to extend the scope of soil moisture observations by hydrological modelling (by increasing spatial and temporal coverage, by enhancing information along the vertical dimension, by reconstructing water fluxes; all of which relating to the specific conditions in Brandenburg with regard to land use, soil, and distance to the groundwater table), and a discussion of practical lessons learned during the establishment and operation of the network.

4. We emphasize, in the new section 4 ("Conclusions and outlook") that the combination of model and data is only one way to make use of this observational network, and that our model application should be seen as a case study rather than any "final" analysis. In order to stimulate future applications in various related fields, and to allow for any interested parties to use the data according to their priorities, we

emphasize once more that the observational and the simulated data is displayed and shared on a public platform (see also section "Data availability") in order to invite and stimulate future collaboration.

5. Even with the aforementioned additions, we aimed to keep the manuscript concise and readable in order to reach a broader audience.

6. Based on the above items, we changed the title of the manuscript to "A new drought monitoring network in the state of Brandenburg (Germany) using cosmic-ray neutron sensing" (removing the "Brief Communication" element).

We hope that the revised manuscript meets your expectations and look forward to your feedback.

**2. Responses to referee #1**

**RC:** *The Authors present a very relevant and interesting initiative to establish an operational ground-based long-term soil moisture monitoring based on cosmic-ray neutron sensors (CRNS). Nine locations and around six months of data are presented. Comparisons with the soil moisture simulated by the agro-hydrological model SWAP are also reported. The manuscript reads well and clear. The preliminary results are meaningful and the discussion fair. Personally, I believe that the main contribution of this initiative is (L39) to bring together a consortium of research institutions and state agencies for establishing a long-term monitoring. And I would congratulate with the Authors for such an effort in moving CRNS research activities towards long term operational monitoring.*

**AR:** Thank you for sharing your positive views.

**RC:** *But back to the manuscript, it reads like an internal report of current status of the initiative and I'm not convinced that is worth a publication. I admit that this bold statement starts from the wish to read novelty in scientific papers and to not consider pure tech-transfer project innovative. In questioning my self-position, I made two actions. First, I look at the description of Brief communications, as it could have provided me with the right angle for judging the manuscript. The description is reported below. In addition, I looked at other published papers with similar vision (to my knowledge). Some of the papers are listed below (Benninga et al., 2018; Cosh et al., 2021). As for the Brief Communications, I leave to the Handling Editor to decide. Anyway, I highlight how the number of pages of the manuscript is way too much. Moreover, its scope might be (c) even if information and data do not seem to be properly disseminated.*

> Brief communications are timely, peer-reviewed, and short (2–4 journal pages). These may be used to (a) report new developments, significant advances, and novel aspects of experimental and theoretical methods and techniques which are relevant for scientific investigations within the journal scope; (b) report/discuss significant matters of policy and perspective related to the science of the journal, including "personal" commentary; (c) disseminate information and data on topical events of significant scientific and/or social interest within the scope of the journal. Brief communications have a maximum of three figures and/or tables, maximum 20 references, and an abstract length not exceeding 100 words. The manuscript title must start with "Brief communication:".

**AR:** We thank the referee for the critical appraisal of the manuscript. Based on the referee comment and the editorial decision, we expanded the scope of the paper, as described in section 1. That way, the paper

became longer, but is not bound anymore to the limitations of the Brief Communication format. We believe that, with the new level of depth, we could also adequately demonstrate the relevance of this study for the NHESS special issue "Current and future water-related risks in the Berlin–Brandenburg region" (see `https://nhess.copernicus.org/articles/special_issue1295.html`) to which our manuscript had been submitted. We are convinced that a new initiative on soil moisture monitoring *in Brandenburg* which is rooted in both research and public administration is worth being reported in such a context.

**RC:** *By looking at other published papers on similar topic (examples above), I have seen much more information in describing the issues and the effort, e.g., to establish such a network, to standardize the observations, to integrate the new data in current platforms, in defining accessibility to end users. Overall, while I'm currently not in favour of the publication of the present manuscript, by considering previous published papers, it might be considered if the Authors put much more effort and they succeed in improving the manuscript, e.g., by sharing their experiences in establishing, maintaining, and managing a fixed environmental sensor network that could be of utility to the community for avoiding mistakes and reproducing good practices.*

AR: Based on the referee comment, we feature, in the new section 2.1, information about the establishment of the network (namely the selection criteria for the monitoring sites), details on the standardization of neutron intensity observations in order to apply the soil moisture estimation (in section 2.2), and a discussion of practical lessons learned during the operation of the network (section 3.4).

**RC:** *Improving metadata (e.g., how was the calibration performed?), transparency (how data have been processed) and data accessibility (e.g., by API-type) should also justify the publication.*

AR: In sections 2.2 and 2.3 of the revised version of the manuscript, we put a stronger emphasis on providing the requested details regarding the estimation of soil moisture from neutron intensities and the set-up and evaluation of our soil hydrological model, as well as the evaluation of both. We also highlighted more clearly that we already provide an openly accessible directory to download both, raw observations *and* soil moisture products, including the required metadata. The directory can be accessed and viewed via a browser interface, and it also allows to obtain static https-links to each of the data files which can be directly used in any automatic data processing environment (which corresponds, in our view, to what the referee refers to as "API-type" access). As already done in the original preprint, we clearly state the availability of this directory in the section on "Data availability".

**3. Responses to referee #2**

**RC:** *In this Brief Communication, the authors present a new initiative in Germany to establish an operational, long-term soil moisture monitoring system based on cosmic-ray neutron sensors (CRNS) for the state of Brandenburg. In general, this is a very positive development, as CRNS has established itself as a new standard method for the continuous determination of soil moisture at the field scale and the widespread deployment of these sensors is very helpful and highly welcome both for environmental research and for the support of water management. In terms of content, this article presents the status of the project as well as a comparison of the measured soil moisture with simulations of the agrohydrological model SWAP. The manuscript is well written [...]*

AR: We thank the referee for the positive feedback on the relevance of our initiative and the readability of the manuscript which are, in our view, important aspects of the evaluation. In the revised version of the manuscript, we tried to maintain the aspect of readability in order to reach an audience beyond the CRNS community.

**RC:** *[...] however, I have to agree with reviewer #1 that the paper reads more like a project report and does not fulfill the NHESS requirements for short communications, especially in terms of the length of the paper.*

**AR:** We appreciate the critical appraisal. As already pointed out above, we expanded the scope of the paper, as described in section 1. That way, the paper became more detailed and extensive, but is not bound anymore to the limitation of the Brief Communication format. We believe that, in the revised version, we could adequately demonstrate the relevance of this study for the NHESS special issue "Current and future water-related risks in the Berlin–Brandenburg region" (see `https://nhess.copernicus.org/articles/special_issue1295.html`)

**RC:** *Also, it may be beneficial to complete the installation and subsequently measure soil moisture over a longer period of time. Once this has been achieved, the authors could publish the results as a data paper.*

**AR:** Our paper is not intended to serve as a replacement of a data paper, although we provide all means to access and use both the raw measurement data and the soil moisture products already at the current stage (see also response to referee 1, and the section on "Data availability" in the manuscript). We appreciate the suggestion that a longer time series could be published as a data publication in the future, but it should be noted that data publications refer to a closed set of data with a fixed DOI which is not what we intend in terms of a continuous long-term monitoring effort. Instead, as pointed out before, we put a stronger emphasis on the analysis of both observational and modelled data.

**RC:** *Finally, we don't learn much from the case study and the context of the comparison of CRNS data with simulation is unclear (see also specific comment further below).*

**AR:** We appreciate the critical appraisal of the case study. We have comprehensively revised the entire manuscript and put more emphasis on how and why CRNS-based soil moisture estimates should be put in context with a soil hydrological model (see also section 1 of this letter).

**RC:** *L38: Why mention only national networks?*

**AR:** In the light of the next referee comment (reg. l. 44), we agree that the CRNS-based soil moisture monitoring network in the context of the ADAPTER project should be mentioned here, too. We added a corresponding sentence which refers to Ney et al. (2021).

**RC:** *L44: Such an initiative is not unique in Germany, see e.g., Ney et al. (2021).*

**AR:** According to our notion the focus of the ADAPTER project was exclusively on cropland while our initiative aims to cover forest and grassland sites, too. Furthermore, we consider the close involvement of various state agencies along the entire process as a distinct feature of our initiative. In order to avoid misunderstandings, we changed the sentence to

> In a transdisciplinary effort, five institutions have combined their resources to establish a CRNS-based network for soil moisture and drought monitoring in Brandenburg [...].

**RC:** *Figure 1: The small maps are hardly recognizable and the situation at the measuring stations is not readable. Also, this information is already available in table 1. Therefore, I suggest removing the small maps.*

**AR:** We thank the referee for the suggestion. The motivation of the maps (including the smaller ones) was *not* to show the landscape attributes at the monitoring locations. This information is, as the referee correctly pointed out, provided in Tab. 1. The idea of the small maps was to give an impression of the overall spatial

distribution of important landscape attributes across Brandenburg. We think that this is relevant with regard to the issues of representativeness and regionalisation. In order to avoid the misunderstanding that the maps should show the attributes at the monitoring locations, we simply removed in the revised manuscript the location markers from the small maps.

**RC:** *L54-71: This sounds very much like an interim report.*

AR: We find it difficult to assess what this statement is supposed to imply. In any case, we find this part very relevant as it lists the selection criteria for the monitoring sites which was also information required by referee #1 (describe issues relevant for the establishment of the network). We hence prefer to keep that part mostly as is.

**RC:** *L116-119: It is not clear to me how this comparison is to be analyzed in this project. What are the consequences if there are discrepancies? Do you then not trust the measurement data, even though the model certainly may have a much higher uncertainty?*

AR: After the fundamental revision of the manuscript, the lines to which the referee referred to do not exist anymore. More importantly, we introduced a new section "Results and discussion" in which we more comprehensively address the comparison of model and observation, including a formal model evaluation by community-accepted metrics and a discussion of discrepancies, as well as a demonstration of how the model could be used to extend the scope of the instrumental monitoring. In section 3.4, we also highlight that the continuous comparison to the model results helped us to detect and fix sensor issues that might otherwise have remained hidden.

**RC:** *L126-129: Describing commitments should not be part of a scientific publication. It would make more sense to present the concrete implementation of a real-time data platform.*

AR: In the revised manuscript, these lines do not exist anymore in the previous form. As for the presentation of a real-time data platform, the amount and complexity of the provided data and products does not yet call for the design of real-time data platforms which would deserve a comprehensive documentation in the context of a journal article. We made a clear statement on data availability in the main text, and, more specifically, in the section on data availability. There, we provide the link to a website that includes a public directory to download both, raw observations *and* soil moisture products. You can access and view that directory via a browser interface, but you can also obtain static https-links to each of the data files which can be directly used in any automatic data processing environment. The directory also contains station metadata in a dedicated table.

**RC:** *L139-140: Formulations such as "some validity" and "some transferability" are too vague.*

AR: We agree. In a revised version of the manuscript, the model evaluation is carried out on a more formal basis, and the corresponding item on "upscaling and transferablility" in the conclusions section was rephrased.

**RC:** *L142-145: This seems to be rather an alternative measurement approach instead of an upscaling approach for the presented CRNS network.*

AR: We agree that the description should be clarified in order to convey a better idea how the railborne CRNS roving could support the upscaling of soil moisture. In the revised version, we hence changed these lines as follows:

[...] an emerging perspective for future research is rail-borne CRNS roving: several locomotives of the Havelländische Eisenbahn AG have recently been equipped with CRNS sensors in order to monitor spatio-temporal soil moisture patterns along selected railway tracks (see Fig. 1). While those of our network locations which are close to these railway tracks (Tab. 1) could be used to verify the spatiotemporal integrity of the railborne data products, the latter could, in turn, be used to validate or train other model-based or data-driven upscaling approaches.

**References**

Benninga, H.-J. F., C. D. U. Carranza, M. Pezij, P. van Santen, M. J. van der Ploeg, D. C. M. Augustijn, and R. van der Velde (2018): The Raam Regional Soil Moisture Monitoring Network in the Netherlands. Earth System Science Data 10, 1, 61-79. https://doi.org/10.5194/essd-10-61-2018.

Cosh, M. H., T. G. Caldwell, C. B. Baker, J. D. Bolten, N. Edwards, P. Goble, H. Hofman, et al. (2021): Developing a Strategy for the National Coordinated Soil Moisture Monitoring Network. Vadose Zone Journal 20, 4, e20139. https://doi.org/10.1002/vzj2.20139.

Heistermann, M., Francke, T., Schrön, M., and Oswald, S. E. (2024): Technical Note: Revisiting the general calibration of cosmic-ray neutron sensors to estimate soil water content, Hydrol. Earth Syst. Sci., 28, 989–1000, https://doi.org/10.5194/hess-28-989-2024.

Ney, P., Köhli, M., Bogena, H., Goergen, K. (2021). CRNS-based monitoring technologies for a weather and climate-resilient agriculture: Realization by the ADAPTER project. In 2021 IEEE International Workshop on Metrology for Agriculture and Forestry (MetroAgriFor) (pp. 203-208). IEEE.

---

## Author Response (AR2)

**Author Response Letter to Editor and Referees**

**Soil moisture monitoring with cosmogenic neutrons: an asset for the development and assessment of soil moisture products in the state of Brandenburg (Germany)**

M. Heistermann, D. Altdorff, T. Francke, M. Schrön, P. M. Grosse, A. Markert, A. Bauriegel, P. Biró, S. Attinger, F. Beyrich, P. Dietrich, R. Eichstädt, P. Grosse, J. Terschlüsen, A. Walz, S. Zacharias, S. Oswald
*NHESS Discussions,* `doi:10.5194/egusphere-2024-3848`
* * *
**EC/RC:** *Editor/referee comment*,    AR: *Author Response*,    ☐ Manuscript text

Dear Dr Somogyvári, dear referees,

we would like to thank both referees for the critical review. Based on the extensive and thorough comments, we fundamentally revised the manuscript. Please find below our detailed point-by-point replies. However, we would like to introduce the main improvements already at this point of the response letter (apologies for the resulting redundancies with the point-by-point replies):

- **Scope and research questions:** As repeatedly emphasized, our original intention had been to briefly introduce – in the context of an NHESS special issue on the Berlin-Brandenburg region – the new CRNS-based soil moisture monitoring network in Brandenburg to the community, and to briefly demonstrate its potential, specifically when being combined with a soil hydrological model. Following, however, the editorial decision after the first round of reviews, we expanded the manuscript to a research paper while still aiming to keep it concise and in line with the original intention. During the second round of reviews, both referees criticized that the main research question remained unclear (referee #3), or that the presented research was not comprehensive enough (referee #4).
  In order to address these concerns, we have now sharpened the aims and research questions, and extended the scope of the paper. Still, we start out from the fact that there is a new CRNS network which allows, for the first time, the possibility to obtain spatially representative root-zone soil moisture estimates across various hydrotopes typical for the state of Brandenburg. In allowing to monitor soil moisture conditions at selected locations, these observations have a value in themselves, and we have now demonstrated this by extending the analysis period and comparing the years 2024 and 2025 with regard to the conditions in spring and summer (as suggested by referee #4). More importantly, yet, these observations provide a new opportunity - a new reference - to assess the validity of soil moisture products for the Brandenburg region - let it be from modelling or remote sensing. That way, we can leverage the observational records to expand the limited temporal, vertical and horizontal coverage of the mere instrumental monitoring, i.e. to use the observational data in order to develop and assess soil moisture products for the state of Brandenburg. This was exemplified in a case study guided by three questions:

  1. How do widely used large-scale soil moisture products (such as ERA5-Land and Soil Water Index of the Copernicus Land Monitoring Service, as suggested by referee #4) capture the observed

soil moisture dynamics in comparison to a soil hydrological model that was set up on the basis of region-specific data?

2. Can the CRNS-based soil moisture estimates help to improve such products, e.g., by means of bias correction?

3. What are the implications of this evaluation for the application of such products in the management of water-related risks in the state of Brandenburg? Which products show the best prospects, depending on the application context?

- **Data paper:** In the course of the process, the possibility of a data paper was repeatedly suggested. However, the format of a data paper does not exist in NHESS, and we have repeatedly argued why a data paper would not be an adequate format for an ongoing monitoring effort. In order, however, to address this issue, we have created a persistent *data publication*, including documentation, that includes the data until September 1, 2025. It is citable, serves a similar purpose as a data paper itself, provides a useful asset to the paper, and a complement to the existing data repository that integrates the real-time observations.

- **Literature cited in the introduction:** Another important concern of referee #3 is the representativeness of the cited literature in the introduction. We apologize for this issue, the root of which lies in our original submission as a ”Brief Communication” for which the total number of references was limited to a maximum of 20. We gladly addressed the referee's corresponding comments which were very helpful.

- **English version of the website:** We have provided an English language version of the website and the data repository (referee #3).

- **Title of the paper:** Referee #4 repeatedly argued that the title does not adequately reflect the content of the paper. We agree. The focus of the paper is not on drought monitoring, but more generally on soil moisture monitoring, and not exclusively on instrumental monitoring, but also on model-based techniques. In order to also adequately express the aforementioned changes in research questions in the title, we changed it to:

> Soil moisture retrieval from cosmogenic neutrons: an asset for the development and assessment of soil moisture products in the state of Brandenburg (Germany)

This new title also puts less emphasis on the network character, as raised by the referee, and more on the application of soil moisture products in the topical context of the NHESS special issue.

- **Network design and representativeness:** Referee #4 also demanded more details on the selection of monitoring locations, and their representativeness for conditions in Brandenburg. In essence, we think that our approach is quite similar to the network design approach followed by Bircher et al. (2012, as cited by the referee) in considering topography (or groundwater depth), land cover, soil type, and climate (precipitation) as well as the availability of additional monitoring infrastructure. The main difference is the number of 30 sensors in the HOBE network which allowed for a more formal approach in sampling across topography, land cover and soil type. For the CRNS network in Brandenburg, site security also played an important role due to the visibility and value of the CRNS stations. In any case, we agree that the criteria should be described more clearly, and that we should also state how well the selected sites actually represent landscape characteristics in Brandenburg. In the revised manuscript, we have hence added more details with regard to the selection criteria and

the representativeness of the selected sites for the environmental conditions in Brandenburg. For that purpose, we also added a new subsection "Study area" in the "Data and methods" section in order to describe the main landscape features in Brandenburg, and their proportions in the area of the state. In the subsequent section "Monitoring network", we then expand how the choice of locations represents a large proportion of Brandenburg's landscape features. We have, however, also emphasized the fact that the ability of a network with eight (or 12) locations to representatively cover a 30,000 km large state such as Brandenburg is inherently limited, an issue that can only be resolved in the future by combining observations with models and remote sensing.

- **Split-sample model calibration/validation:** In response to referee #4, we have implemented a split-sample calibration/validation for the SWAP model (2024 data for calibration, 2025 data for validation). We also implemented a bias-correction for the competing large-scale soil moisture products and followed the same split-sampling approach as for the SWAP model.

While we are grateful for the constructive comments, we would also like to maintain that, in the process, the number and extent of comments has accumulated to a considerable level. We understand that there will always be additional and useful ideas of what could be done. But while we are open to extending the scope of the paper, we also need to make sure to constrain it to a manageable level, even if that means that not all suggestions can implemented.

As mentioned above, you will find point-by-point replies to all comments on the following pages.

We hope that we have, this way, addressed the referees' concerns. Thanks again for your feedback and your support of this process.

Kind regards,
Maik Heistermann
(on behalf of the author team)

**1. Responses to referee #3 (report #1)**

RC: *[1.1] [...] What is the main research question behind it? Does that network mainly serve as a number of reference points for drought monitoring inferred from railroad measurements? If so, why is there no analysis presented on that? Or do the authors intend to provide data on hydrologically relevant locations within Brandenburg in order to (later) feed that into a larger scale model? What role does clustering of sensors have in that context and to which extend is the scale gap discussion relevant?*

AR: In order to address the referee's comment, we have sharpened the aims and research questions, and extended the scope of the paper. Still, we start out from the fact that there is a new CRNS network which allows, for the first time, the possibility to obtain spatially representative root-zone soil moisture estimates across various hydrotopes typical for the state of Brandenburg. In allowing to monitor soil moisture conditions at selected locations, these observations have a value in themselves, and we have now demonstrated this by extending the analysis period and comparing the years 2024 and 2025 with regard to the conditions in spring and summer (as suggested by referee #4). More importantly, yet, these observation provide a new opportunity - a new reference - to assess the validity of soil moisture products for the Brandenburg region - be it from

modelling or remote sensing. That way, we could leverage the observational records to expand the limited temporal, vertical and horizontal coverage of the mere instrumental monitoring, i.e. to use the observational data in order to develop and assess soil moisture products specifically for the state of Brandenburg. This will be exemplified in a case study that is guided by three questions: (1) How do widely used large-scale soil moisture products (such as ERA5-Land and the Copernicus Land Soil Water Index, as suggested by referee #4) capture the observed soil moisture dynamics in comparison to a soil hydrological model that was set up on the basis of region-specific data? (2) Can the CRNS-based soil moisture estimates help to improve such products, e.g., by means of bias correction? (3) What are the implications of this evaluation for the management of water-related risks in the state of Brandenburg? Which products show the best prospects, depending on the application context?

As to the referee's question regarding the rail-based CRNS measurements, we would like to clarify that these are not a subject of this study. In the manuscript, we mentioned the rail-based CRNS measurements for two reasons: first, because the proximity to rail-tracks was one (though rather subordinate) criterion in the selection of monitoring locations, and, second, because the rail-based CRNS measurements provide a new future perspective for spatial upscaling. We have clarified these two aspects in the revised manuscript.

The scale gap discussion was removed from the introduction.

**RC:** *[1.2] The provided data seems to be available only in German (online interface). Please provide an English version.*

AR: In the course of the revision, we have developed a bilingual (English, German) version of the website. The "data" tab on the website merely provides a link to the actual data repository. Within that data repository, all metadata and readme files that are required to interpret the actual data files had already been provided in English.

**RC:** *[1.3] The introduction unfortunately suffers from unrepresentative selection of literature with the focus on self-citations.*

AR: We agree. The reason for this is that in the original submission as a "Brief Communication", the number of references was limited to a maximum of 20. So we had to be very strict with citations in the introduction. As we stated in our previous response to the editor, our ambition with the revised manuscript was still to be brief and concise, which partly came at the cost of comprehensiveness. We did not intend to inappropriately advertise our own papers. Certainly, we take the referee's concern very seriously and have revised the referencing in the introduction, considering also the referee's following comments.

**RC:** *[1.4] In the paragraph of soil moisture monitoring and its data sources Oswald et al. 2024, Babaeian et al. 2019 and Schmidt et al. 2024 are cited. With Babaeian et al. 2019 presenting a more fundamental overview, Oswald et al. 2024 seems to be not strongly linked to the respective claims in the text. There is no reference to point scale probes which make up the largest amount of data sources. Schmidt et al. 2024 is not representative for remote sensing. To make a more specific example: Oswald et al. 2024 is cited for the following claims: A: "soil moisture monitoring is widely acknowledged", B: "remote sensing products are limited by shallow penetration depths, low overpass frequencies, and vegetation-related uncertainties", C: "Hydrological models have the potential to overcome such limitations". A is not within the scope of Oswald et al. 2024. To the contrary, this topic is covered in literature since 50+ years. C is neither very well covered as Oswald et al. 2024 focuses on ML applications, whereas here land-surface models for example should be cited. Oswald et al. 2024 provides a short overview about remote sensing product and therefore the claim is covered but again, here the authors should focus on primary sources instead of yet again cite their own works.*

AR: Thanks for the detailed investigation. We have replaced most references to Oswald et al. (2024) with a more extended list of primary references. We also extended the citation list around Schmidt et al. (2024), while it is highly relevant in this context, being one of the first studies who comprehensively compared CRNS with multiple remote sensing products across Germany. Throughout the paper, we also sought to add references in a more differentiated way to make clear which claim they are to support.

RC: *[1.5] In the paragraph for CRNS networks Heistermann et al. (2023), Ney et al. 2021, Zacharias et al. 2024 are cited - all publications with one of the authors or affiliated German groups. The authors mention the larger networks only in one sentence without citation "Only few countries have already established long-term CRNS monitoring networks at the national scale (e.g., the USA, UK).", even leaving out CosmOz Australia.*

AR: In the revised manuscript, we have provided a more comprehensive representation of previous efforts on long-term CRNS monitoring networks at the national scale, including CosmOz Australia.

RC: *[1.6] The section about CRNS does not cite the relevant literature - neither does Andreasen et al. 2017 explain the method, nor is Schrön et al. 2017 representative for the footprint. Likewise, Altdorff et al. 2023 is neither the first nor the most relevant publication for roving. To conclude, the introduction is, for a research paper, fragmentary, unrepresentative and too much narrowed down to the publications of the researchers themselves.*

AR: Due to the limitation of citations in the earlier brief communication format, we used Andreasen et al. (2017) as a general reference where readers could find references therein which explain the CRNS method in more detail, for instance. However, we agree that in the current format the reference is not sufficient, and added further citations from Zreda et al. (2008, 2012). We disagree, though, that Schrön et al. (2017) is not representative for the footprint: the vertical and horizontal weighting functions described there are still the state-of-the-art and de-facto community standard. However, we agree that providing just one number (here 10 ha) with regard to the footprint size does not sufficiently account for the strong dependency on soil water content. In the revised version, we have changed the specification from area (ha) to radius (m) which ranges from 130–240 m based on Köhli et al. (2015) and then Schrön et al. (2017). With regard to Altdorff et al. (2023), we fully agree that this reference is not comprehensive and representative for CRNS roving, but similar to the argumentation above, one of the most recent studies which addressed CRNS roving and provide references therein to earlier work. In the course of the revision of the introduction, we have added a brief introduction to CRNS roving to motivate the location of the CRNS sites near rail networks, and we added more representative but also recent citations to support that argument (e.g., Franz et al., 2015; McJannet et al., 2017; Schrön et al., 2021; Handwerker et al., 2025)

RC: *[1.7] l41: "So far, however, CRNS has mainly been used in experimental contexts" - what does experimental (short-term) contexts mean? If ignoring other literature about CRNS networks, the reader might come to the conclusion that probes would be installed on a short term basis, but as a matter of fact a large number of the probes are used in long-term monitoring contexts, so that claim seems to be not very well supported.*

AR: We agree, and have deleted that sentence.

RC: *[1.8] l47: "The first attempt in Germany to systematically apply CRNS technology at the federal state level was initiated in 2024". The authors previously cited Ney which is definitely an application of "CRNS technology at the federal state level prior to the authors' initiative.*

AR: In the previous sentence of the manuscript, we explained that the ADAPTER network to which Ney et al. (2021) refer does not aim at the entire federal state of North Rhine-Westphalia, and that it only covers cropland sites. That is why we, despite the (as of now) still limited number of sensors and the lack of coverage in

southern Brandenburg, consider the effort in Brandenburg to be different. Then again, we do not intend to unnecessarily insist on the uniqueness of the Brandenburg network, as our main point is its relevance in the Brandenburg context (which is also the context of the special issue in NHESS to which this manuscript was submitted). Therefore, we changed the sentence to

> For the federal state of Brandenburg, a consortium of research institutions and state agencies was formed in 2024 in order to implement and maintain a CRNS-based soil moisture monitoring network, designed to represent typical combinations of land use, soil and groundwater conditions in the state.

**RC:** *[1.9] l77: "Such a station includes a neutron detector, logger and telemetry, solar power supply, and sensors for barometric pressure, temperature, and humidity, as well as conventional point-scale sensors of soil moisture in various measurement depths as an additional reference" - please name sensor type and manufacturer.*

AR: We have added manufacturer and sensor type for the conventional point-scale sensors of soil moisture, and also added the manufacturers of the different CRNS systems in the caption of Tab. 1.

**RC:** *[1.10] l85: "Sensor footprint that are homogeneous with regard to these attributes are preferable with regard to the interpretation of the CRNS signal." - this statement seems to conflict with the approach to collocate sensor with (considerably dry) train lines. Altdorff et al. 2023 shows in Fig. 1 a sensor in a distance of approximately 1 m to trainlines, which would make the sensor potentially susceptible to the road effect. How distant are these sensors to the rail infrastructure and in which way did the authors account for that?*

AR: Thank you for this comment which is quite to the point. First of all, we would like to emphasize that the CRNS sensor shown in Fig. 1 of Altdorff et al. (2023) is located in a different federal state and is not part of the Brandenburg monitoring network. For those locations in the Brandenburg network which are close to railroads, we still had to face a trade-off between proximity to the railroad and homogeneity of the CRNS footprint (as we stated in the paper, homogeneity was *preferable*, but not an exclusive criterion). In order to limit the (rail-)road effect on the CRNS signal, we kept a distance of 10 meters to the rail tracks for the CRNS stations in Golm (GOL) and Paulinenaue (PAU). For the sensor in Marquardt (MQ), the distance is larger than 100 meters. So while for GOL and PAU, there might still be some minor effect of the trainlines on the neutron signal, we estimate that the overall contribution to the signal is less than 5 percent. Future analysis will however keep an eye on these influences and will acknowledge them in signal interpretation and potential specific corrections.

**RC:** *[1.11] Table 1: "Land use from OpenStreetMap contributors (2024)" - why do the authors cite the (relatively vague) OSM maps. Did they not characterize the sites themselves in terms of land use and soil texture? That seems to be a minor red flag.*

AR: We did characterise the land use ourselves, and the reference to OSM in the table is in fact redundant and was removed. Yet, we have to disagree with regard to the "vagueness" of the OSM maps. These are usually very precise, at least for Brandenburg, and a fully valid source of local scale land use mapping. We also have contributed to the OSM-data wherever this was not the case.

**RC:** *[1.12] l111: "The sensitivity of the neutron detector relative to a known reference" - in what way is that procedure a 'general calibration'. If each sensor still has its own normalization factor this procedure seems to be not different than as has been done at other sites.*

AR: We would like to refer to Heistermann et al. (2024) for the details. The main point is, however, that this

"normalisation factor" is not obtained by means of calibrating an $N_0$ value (or other kind of normalisation factor) on observed soil moisture data, but by independently measuring the sensitivity of the detector via collocating it with another detector. That way, we independently quantify the effect of detector sensitivity. The idea of the "general calibration" is hence not to ignore the local effects on the CRNS signal (other than soil moisture), but to quantify them independently and not rely a local calibration based on local soil moisture observations (since these are inherently uncertain due a number of reasons).

**RC:** *[1.13] l114: "The spatial variation of incoming cosmic radiation was accounted for by using the PARMA model (Sato, 2015)" - what does that mean? Did the authors conduct an own analysis and how do they map the high-altitude site JUNG to their data?*

AR: Again, we would like to refer to Heistermann et al. (2024) for a detailed explanation. The main point, though, is that we account for the temporal variation of the incoming neutron signal by using the observations at Jungfraujoch (JUNG), while we account for the *spatial* variation of the neutron signal by means of the PARMA model (Sato, 2015).

**RC:** *[1.14] l122: "we followed the weighting procedure outlined by Schrön et al. (2017)" - did the authors use the sampling scheme as well or the footprint function?*

AR: We used Schrön et al. (2017) in order to obtain the vertical and horizontal weights, but not for the sampling scheme. We revised the manuscript to clarify this.

**RC:** *[1.15] l211: "At this point, we should reiterate that the focus of this paper is to introduce the new soil moisture monitoring network in Brandenburg" - the introduction of a monitoring network by itself is not a research goal. The authors claimed to try to focus on providing a substantial contribution, not an outlook based on preliminary findings.*

AR: We have removed the statement, also in the light of the newly formulated research questions. We agree that the introduction of a monitoring network is not a research goal in itself.

Still, we consider the monitoring effort itself as highly relevant to the audience of the NHESS special issue, i.e. to the research community that is concerned with topics such as drought monitoring, hydrological modelling, water resources management and climate change adaptation in the Berlin-Brandenburg region.

**RC:** *[1.16] l214: "we maintain that the model is able to reproduce the observed soil moisture dynamics in the upper 30–50 cm of the soil"- The authors previously mentioned that the sites are only capable of measuring the soil water content to a depth of 30 cm. Explain this statement.*

AR: We apologize, but we could not find such a statement in the previous version of the manuscript, although the typical penetration depth of CRNS measurements is in fact often referred to as 30 cm for average conditions. As shown in Fig. 2 (original and revised), the penetration depth depends much on the soil moisture, and varies roughly between 25 and 60 cm. Please also note that the corresponding paragraph does no longer exist in the same form in the revised manuscript, yet we have clarified in the text that the penetration depth varies in time.

**RC:** *[1.17] Fig. 2: much too small, please enlarge.*

AR: For the revised figure, we have used a figure layout with four rows and two columns in order to allow for a larger representation of the figure subpanels.

**RC:** *[1.18] l311: "ware" -> "were"*

AR: Was changed to "will" in the revised version.

**RC:** *[1.19] in the conclusions section "User-oriented monitoring products" - many research questions are presented which are not covered in the manuscript.*

**AR:** We would like to emphasize that the section is entitled "Conclusions and outlook", exactly because we would like to use the opportunity to outline prospective lines of research and applications. We think that this is a common and well-accepted scope for such a section. However, we think that with the revised version of the manuscript, the points that are addressed in the outlook now align much better with the subject of the paper.

**RC:** *[1.20] Additionally, with only a German data interface most of the potential users face a language barrier.*

**AR:** As pointed out above, we have also provided an English version of the website, and also of the required files in the data repository.

**RC:** *[1.21] l350: "In any case, (...)" - that is a rather colloquial statement and potentially does not contribute in a meaningful way.*

**AR:** We have removed "In any case" from the statement.

**RC:** *[1.22] While the contributions statement suggests MH, TF, DA, AB, AM, and SO were the core researchers, the other co-authors may have had minor roles, potentially indicating 'gift' or 'coercive' authorships.*

**AR:** The authorships are neither "gift" nor "coercive" authorships. Yet, we agree that the contributions should be documented in more detail, and have revised the manuscript accordingly. We followed the CRediT contributor roles taxonomy (link), as also recommended by NHESS guidelines for manuscript composition.

**2. Responses to referee #4 (report #2)**

**RC:** *[2.1] The paper is interesting as it presents a soil moisture network based on CRNS. The data will be valuable for multiple research applications, and thus it is of interest to publish this manuscript. Looking at the previous reviews I agree it is too much for a Brief communication, but for a research paper it is not innovative or comprehensive enough. It is more suitable as a data paper. For a research paper, the manuscript needs significant improvements before publication.*

**AR:** We understand the concern, and have increased our efforts to strengthen the research component of the manuscript (see also comment [1.1].

We would also like to comment on the idea of a data paper. This idea had already been suggested in the first round of review. We think that this paper is not and should not be a data paper for mainly two reasons: first, a data paper typically refers to a snapshot of data, even more typically after the "end" of a process of data collection (a campaign, a project, ...) that yielded a substantial amount of data. Here, however, our original and still valid motivation is to point the community to an ongoing activity in order to involve researchers and users as early as possible. Technically speaking, we do not want to refer to a closed dataset, but to a continuous data collection effort, and to a network that is still evolving. Second, NHESS does not provide the format of a data paper, while we are (and were, from the very beginning) convinced that this paper fits very well into the context of the NHESS special issue "Current and future water-related risks in the Berlin–Brandenburg region".

As a solution, we would like to suggest a compromise: we have published the dataset in its state on September 1, 2025, in a dedicated data publication portal that supports the FAIR principles, including an adequate documentation and a persistent DOI (10.23728/b2share.dfde74f4be294bd7b927f67988365f8e). We have

added this data publication as an "asset" to this paper, and we refer to it in the revised manuscript. That way, we have linked the paper to a published dataset, while the scope of the paper is still beyond that of a data paper. At the same time, we maintain the website and data portal to allow for an ongoing provision and presentation of data with short latency.

We hope that this solution meets the demands expressed by the referee.

**RC:** *[2.2] In general, the title points towards a drought monitoring network which is installed in Brandenburg. My expectation is then, that there is a detailed description of the network set-up, how locations were chosen, a thorough validation of the network and with a focus on its ability to monitor drought. However, this is not sufficiently represented in the manuscript. The selection of locations is not satisfactorily described, the validation and quality assurance is lacking (a validation based on one observation per station is not sufficient), and there is no mention of drought monitoring. This needs to be improved in the manuscript before this can be published as an introduction to a new monitoring network in the form of a research paper.*

**AR:** This comment raises quite a variety of issues, and we will try to address them one by one.

- **Title**: We agree that the previous title did not sufficiently reflect the content of the paper. Specifically, the focus of the paper is not exclusively on drought monitoring, but more generally on soil moisture monitoring, and not only on instrumental monitoring, but also on model-based techniques (and now also remote sensing products). In order to also adequately express the aforementioned changes in research questions in the title, we changed it to: "Soil moisture monitoring with cosmogenic neutrons: an asset for the development and assessment of soil moisture products in the state of Brandenburg (Germany)" This new title also puts less emphasis on the network character, as also raised by the referee, and more on the applicability of soil moisture products in the topical context of the NHESS special issue.

- **Network design and selection of locations**: As pointed out in the previous item, we revised the title to remove the emphasis on "drought monitoring network" in order to avoid raising false expectations with regard to the aspects of "network" and "drought". Regarding the choice of monitoring locations, we think that our approach is quite similar to the network design approach followed by Bircher et al. (2012, as cited by the referee) in considering topography (or groundwater depth), land cover, soil type, and climate (precipitation) as well as the availability of additional monitoring infrastructure. The main difference is the number of 30 sensors in the HOBE network which allowed for a more formal approach in sampling across topography, land cover and soil type. For the CRNS network in Brandenburg, site security also played an important role due to the visibility and value of the CRNS stations. In any case, we agree that the criteria should be described more clearly, and that we should also state how well the selected sites actually represent landscape characteristics in Brandenburg. In the revised manuscript, we have hence added more details with regard to the selection criteria and the representativeness of the selected sites for the environmental conditions in Brandenburg. For that purpose, we also added a new subsection "Study area" in the "Data and methods" section in order to describe the main landscape features in Brandenburg, and their proportions in the area of the state. In the subsequent section "Monitoring network", we then expand how the choice of locations represents a large proportion of Brandenburg's landscape features. We have, however, also emphasized the fact that the ability of a network with eight (or 12) locations to representatively cover a 30,000 km large state such as Brandenburg is inherently limited, an issue that can only be resolved in the future by combining observations with models and remote sensing.

- **Validation and quality assurance**: The referee stated that "the validation and quality assurance is

lacking (a validation based on one observation per station is not sufficient)". We assume that the comment refers to the validation of the CRNS-based soil moisture retrieval, and we disagree with that notion. The general calibration approach that was used in our study to estimate soil moisture from neutron counts, as documented by Heistermann et al. (2024), is based on a large sample of 75 sites across Europe (including Brandenburg). While we think that it is interesting to compare our CRNS-based estimates to the references from the sampling campaigns, it is not common procedure in the CRNS literature to carry out such a validation. Given the number of sites, we think that our study is in fact exemplary in providing an independent assessment of the soil moisture estimation at all, while other studies often use the groundtruthing data only for calibration, and not for validation. In response the the referee's comment, we have, however, discussed in more detail the uncertainties that result from the limited number of independent reference measurements.

- **Drought monitoring**: As pointed out above, we have removed the emphasis on drought monitoring from the title of the manuscript which we hope addresses the issues raised in this comment. However, in the new section "Implications for the management of water-related risks in Brandenburg", we discuss aspects related to drought monitoring and drought risk management, too.

**RC:** *[2.3] I realize that the network is only operational since spring 2024, but currently this gives you one year of data. In addition, the current situation in Brandenburg is pressing with a drought warning and high alert levels (European Drought Observatory), whereas spring 2024 was more humid. So my suggestion is to extend the analysis period to include a full year of data including the last months to also observe the onset of drought conditions in Brandenburg in 2025. One option is for example to calculate a drought indicator based on soil parameters, i.e. the Soil Water Deficit Index, which allows you to also showcase the drought monitoring capabilities.*

**AR:** We thank the referee for this comment, and appreciate the suggestion of the "Soil Moisture Deficit Index" (SMDI, as published by Narasimhan and Srinivasan, 2005). Yet, it illustrates the challenge that we face with the relatively short (approx. one year) observational time series of our CRNS-based soil moisture (as also pointed out by the referee): the SMDI requires long-term soil moisture statistics (minimum, median, maximum) which can only be obtained by model-based extrapolation in time - as already demonstrated in the previous manuscript version. Hence it cannot be applied to the observational series.

Yet, we agree with the referee that it makes sense to extend the analysis period to include spring and summer of 2025, as these in fact contrast well with the conditions in 2024. Given the changes in the title and the scope of the paper, though, we focused on soil moisture instead of drought indices. In the revised section 3.1, we now briefly discuss the differences between 2024 and 2025 with regard to the presence of very dry conditions.

**RC:** *[2.4] Furthermore, there are many applications of soil moisture data such as reconstruction of water fluxes and enhancing information along the vertical dimension, but these are all described very minimal without proper evaluation of the results. In addition, they are all based on the modeled soil moisture and not CRNS soil moisture. The paper seems to have a hard time balancing between the CRNS data and the model, and the current content of the paper does not reflect the title of the paper.*

**AR:** We agree that it was challenging to find an adequate balance between the presentation and analysis of observations and model results. This is because, on the one hand, the observations represent a means in itself, as they provide direct and independent information on the soil moisture status across the federal state of Brandenburg. On the other hand, the use of a soil hydrological model unlocks a variety of additional applications, namely by extending the scope of analysis in space and time (see comment [2.3]), and from soil moisture to vertical water fluxes. This kind of extended view requires a validated model, and the observations are one component for such a validation. So yes, it is difficult to find a balance between the adequate

introduction of the observational network, and the demonstration of its combination with a model. The latter remains, in the context of this manuscript, admittedly superficial. Still, we are convinced that this demonstration is an integral component in providing a more comprehensive perspective on this effort. We have revised the paper accordingly to make this even clearer, revised the research question (see comment [1.1]), and also adjusted the title accordingly (see [2.1]).

RC: *[2.5] line 47 - 50: Adjust this, because this statement is not entirely true, CRNS data is available in the International Soil Moisture Network and also available through other sources.*

AR: We are not sure if the referee in fact refers to ll. 47 to 50 of the manuscript under revision. We assume the reference is rather to the preceding lines about other CRNS network efforts. Also in response to the comments of referee #3, we have revised this section to adequately account for CRNS networks established previously, including CRNS data available through the International Soil Moisture Network.

RC: *[2.6] line 94: Are these stations already installed now? Otherwise I would change this to summer.*

AR: The referee is correct (we assume the reference is to l. 81 of the manuscript version under revision), and, indeed, touches a sore spot: the new locations are fixed and the corresponding CRNS stations are purchased and ready to be deployed - except that the remote data transfer is not yet fully functional. This is the first time that we experience such a persistent issue: for more than four months now, our technicians, together with the manufacturer, are trying to resolve it, and, in fact, there has been considerable progress in the past weeks. Yet, we will only deploy the stations in the new monitoring locations when remote data transfer has been robust over a test period of four weeks from our test field site near Potsdam. Given that the recent firmware fixes are successful, the sensor deployment is, of now, planned for November 2025. Updates will be provided via the news page of our website (`https://cosmic-sense.github.io/brandenburg/en/aktuelles/`). In the revised manuscript, we have changed the information across the various occurrences in the paper to a designated instrumentation date in November 2025.

RC: *[2.7] The Data and Methods section is missing quite some information, and this should be added to the manuscript. Some methods are mentioned but not explained. In detail the following points should be added to increase the quality of the manuscript: What has been the method for selecting the sites? A representative network is important, as is written in the introduction, but there is no description on how the sites were chosen based on land cover, terrain and soil characteristics. Looking at the maps in Fig. 1 it seems that the south of Brandenburg is not represented, which also includes some areas with a low groundwater table. When planning a network usually one takes an approach to make sure you cover all typical landscapes (or hydrotopes as Bircher et al did for the HOBE network).*

AR: This comment refers to the issue of network design and selection of locations. We hence kindly refer to our response to comment [2.2].

RC: *[2.8] In addition, please provide an overview how the locations and characteristics of the stations (table 1) correspond to the characteristics of Brandenburg, are you really covering everything? This is especially important for drought monitoring, which is one of the main purposes of this network. In addition, if in the future the ministries or other institutions are going to use this network for drought monitoring, or if this even is going to be used for agricultural applications (i.e. drought impact analysis, insurance purposes) the representativeness of the network is crucial.*

AR: Again, this comment raises the issue of network design and the resulting representativeness, so please also refer to our response to comment [2.2]. Based on the referee's suggestion, we have added more details with regard to the representativeness of the selected sites for the environmental conditions in Brandenburg. For that purpose, we also added a new subsection "Study area" in the "Data and methods" section in order to

describe the main landscape features in Brandenburg, and their proportions in the area of the state. In the subsequent section "Monitoring network", we then expand how the choice of locations represents a large proportion of Brandenburg's landscape features.

Yet, we would like to emphasize that we do not and did not claim "to cover everything". In fact, we had already emphasized the limited ability of a network with eight (or 12) locations to representatively cover a 30,000 km large state such as Brandenburg, an issue that can only be resolved in the future by combining observations with models and remote sensing. While we have included this issue in the research questions of the revised manuscript (see response to 1.1), it remains a major challenge for prospective research, as also outlined in the section on conclusions and outlook.

**RC:** *[2.9] Also, I suggest to add the stations that are/will be installed in spring 2025 for completeness to table 1.*

AR: We have added the stations to Tab. 1, including a row header to indicate the dedicated date of installation (November 2025, please also see out response to comment [2.6]).

**RC:** *[2.10] line 120: How were the four soil sampling locations within the CRNS footprint chosen? How do they represent the footprint characteristics?*

AR: The sampling locations were chosen randomly within the near range of the footprint (i.e. within a radius of 20 m), given that this near range exerts the strongest impact on the signal. Certainly, the limited number of samples for bulk density is one source of uncertainty. We have discussed the resulting uncertainties in more detail in section 3.1 of the revised manuscript.

**RC:** *[2.11] In line 79 it is mentioned that there are conventional point-scale sensors of soil moisture installed at the site. But in line 129-130 you mention that "To evaluate the CRNS-based soil moisture estimates ($\theta_{CRNS}$), reference observations within each CRNS footprint were obtained from the aforementioned soil sampling campaigns". From the results it seems only the sampling is used for validation. But why did you not also use the point-scale sensors for validation. Of course, spatial representativeness can be a point of concern, but a validation on a single sampling date is also not representative or a proper validation. I suggest to add also a validation over time, i.e. with the point-scale soil moisture sensor data.*

AR: It is correct that we only use the soil moisture observations from the sampling campaign to obtain reference estimates which we then use to assess the CRNS-based soil moisture estimates. Having more additional sampling campaigns in order to obtain more reference values was not feasible based on the available resources. However, we would like to emphasize once more that our soil moisture estimation follows the "general calibration" procedure as documented by Heistermann et al. (2024) which is based and evaluated on a large sample of 75 sites across Europe. We have pointed this out more clearly in the revised manuscript (section 2.3). Still, we think that it is, in fact, good practice to compare our CRNS-based estimates to the references from the sampling campaigns, as we did, and it is not common to carry out such a comparison with independent measurements in CRNS applications (see also our response to comment [2.2]).

Regarding the installed point-scale sensors, these do only constitute one single profile with measurement depths at 5, 10, 20 and 30 cm. Using such a single profile as a validation reference would be pointless, given that it is exactly the lack of representativeness of such point-scale sensor which we aim to overcome by using the CRNS technology. Neither an agreement nor a disagreement of our CRNS-based estimates with a single profile would be conclusive with regard to the validity of the CRNS-based time series. This is why we prefer not to implement this suggestion.

**RC:** *[2.12] Line 135, please write out the method you used to get the weighted reference soil moisture, not just the publication. And also, which method and metrics did you use to then validate your CRNS soil*

*moisture?*

AR: We have revised the manuscript in order to explain the weighting procedure in more detail, and to state which metrics have been used for the evaluation: the error (difference between $\theta_{CRNS}$ and $\theta_{REF}$) at each location, and the mean error (ME) and the root mean squared error (RMSE) across all eight monitoring locations.

RC: *[2.13] Table 2 contains many acronyms that are not explained, please write them out in full.*

AR: We assume that the referee refers to the first column with the parameter names. These parameter names are listed only to provide an exact reference to the model parameters. The meaning of these parameters is explained in the column "meaning" which provides, in our view, sufficient information to understand the parameter's relevance. For any further detail, we refer to the model documentation (Kroes et al., 2017), which we clarified in the revised version of the manuscript.

RC: *[2.14] Section 2.3 what is the spatial and temporal resolution of your model?*

AR: The temporal resolution is one day. The spatial resolution (i.e. in the vertical, as it is a 1-dimensional model) is 1 cm (at 0-5 cm), 2.5 cm (at 5-15 cm), then 5 cm (at 15-50 cm), 10 cm (at 50-100 cm), 10 cm (at 100-200 cm), 20 cm (at 200-500 cm) and 50 cm below. The actual depth of the soil column depends on the depth of the groundwater surface at the corresponding location. We have added the corresponding information in the revised manuscript.

RC: *[2.15] For the inputs to the pedotransfer function, why didn't you use the actual soil information from the soil sampling campaign and instead did a so-called "fine-tuning"? Is soil texture (sand, silt, clay) data not available for the sites from your sampling campaign? It would be beneficial to use this instead of tweaking your model inputs with estimated sand, silt and clay values to a point to get satisfactory results. If it is not available I suggest to at least check your calibrated sand, silt and clay content with the bulk density values from your samples to see if this is comparable.*

AR: We use the texture data from the soil map (BUEK300) as input to the pedotransfer function because this information is available anywhere in Brandenburg. This is a precondition for any spatial upscaling for which we would like to provide a perspective on the basis of our analysis. We have pointed this out more clearly in the revised manuscript. We do not consider this as "tweaking" because we keep the fractions of sand, silt and clay within the ranges specified by the soil map. Given the generally high sand content, we do not agree, either, that comparing the calibrated sand, silt and clay contents to the bulk density at the measurement locations would be helpful since bulk density will largely be influenced by soil structure, as a result from, e.g., vegetation and agricultural management practices and organic matter – aspects which, however, are beyond the scope of this study. Please also note that, following the comprehensive revision of the manuscript, we now evaluate the uncalibrated version of the model as well as a calibrated (fine-tuned sand content) model (see section 2.4.1 and 2.5 of the revised manuscript).

RC: *[2.16] Section 3.1 gives a short overview of the soil moisture estimation and validation. First, which in situ soil moisture measurements were now used here, the stations or the samples?*

AR: As pointed out in our response to comment [2.11], we used the soil moisture obtained in the sampling campaigns. We have clarified this in the manuscript in section 2.3.

RC: *[2.17] I suggest to also use the RMSE as a quality control metric, as this is quite common in soil moisture validation studies (i.e. Gruber et al. 2020 for satellite soil moisture retrievals).*

AR: We have replaced the MAE with the RMSE as a metric.

**RC:** *[2.18] Also, I suggest to add some more references to publications that have validated CRNS sensors, to place your network in context to other CRNS networks, i.e. what were the metrics obtained for the TERENO network.*

A fair validation is only possible for sites with a number of point-scale measurements that is able to sufficiently represent the footprint area around the CRNS station, and which have not been used for the CRNS calibration. Usually, any such measurements are used for calibration, so that an independent validation is rarely conducted. Still, few studies carried out such a validation, either based on additional sampling campaigns or based on continuous sensor networks: Cooper et al. (2021) stated that "repeat calibrations using secondary samples have been conducted at two COSMOS-UK sites to explore the accuracy of the derived VWC obtained on a particular day [...]. There was below $0.03\,\mathrm{m^3\,m^{-3}}$ difference in volumetric water content". Coopersmith et al. (2014) used an in-situ network at one COSMOS station for validation and found the RMSE "well below $0.04\,\mathrm{m^3\,m^{-3}}$ [...]". Schrön et al. (2017) followed a similar approach and found RMSE values between 0.006 and $0.051\,\mathrm{m^3\,m^{-3}}$ across four CRNS sites in Germany, three of which belong to the TERENO program. Finally, Iwema et al. (2015) systematically validated the effect of the number of calibration measurements at two TERENO sites in Germany and found validation MAE values between about 0.04 and $0.07\,\mathrm{m^3\,m^{-3}}$ (depending on the number of calibration dates from one to six). We consider the validation results obtained in the context of our study quite well in line with these references, and we have added the corresponding references and results to our manuscript in the context of discussing the errors estimated for the sites in our network in Brandenburg. At the same time, we would like to emphasize that great care has to be taken when such error metrics are compared across sites or networks. Currently, there is work in progress to quantify such metrics for CRNS sites across the globe, and we appreciate the reviewers confirmation that such research would be valuable.

**RC:** *[2.19] Especially, as the soil moisture values are very low at the time of sampling, your MAE and bias might be not representative (i.e. higher errors might occur with a higher variability in soil moisture).*

**AR:** We do not agree with the general statement that "soil moisture values are very low at the time of sampling". Given the soil conditions Brandenburg, this only applies to the locations Booßen and Kienhorst.

**RC:** *[2.20] One of my major concerns is that you have calibrated your model parameters on the actual CRNS SM data. Then you use the same datasets, both CRNS and model SM to evaluate your results. This is not a proper way of doing an evaluation, as you have used the same data for calibration and validation, making these results invalid. So, either split your data in training and test (cal/val) data to asses model performance, or use another approach to obtain the model parameters.*

**AR:** Thanks for this comment. For the revised manuscript, we have implemented a split-sample calibration/validation for the SWAP model (2024 data for calibration, 2025 data for validation). We also implemented a bias-correction for the competing large-scale soil moisture products and followed the same split-sampling approach as for the SWAP model. Please note that the overall presentation of results is now fundamentally different since the validation metrics for both the uncalibrated and the calibrated SWAP model are now presented in section 3.2 together with the metrics for the large-scale soil moisture products (without and with bias correction).

**RC:** *[2.21] And, to assess your model performance, why not use the point-scale SM measurements as independent reference?*

**AR:** Please refer to our response to the referee's comment [2.11].

**RC:** *[2.22] In section 3.2 please add a paragraph on how your calibrated parameters compare to your soil samples. From Table 4 it can already be observed that for the station OEH your calibrated sand, silt and*

*clay is at the boundary conditions. This means your model had difficulties for this site. Please check this.*

AR: It is correct that the fine-tuned sand content is at the upper end of the range specified by the soil map for the station OEH which implies that the model does not fully succeed in producing the drier conditions as observed the the CRNS (overestimation of soil moisture). It is exactly why we only fine-tuned the sand content within the limits provided by the soil map. However, the reasons for the overestimation remain unknown - they could be caused by uncertainties of the soil map, the model, but also by a systematic bias of the observations. Please note that soil texture values were not obtained for all sites.

RC: *[2.23] Also, in table 4, I assume T it Ton, but change this to C for clay, and also change U.*

AR: Thanks for pointing this out. The letters in fact represented the German abbreviations. We have changed "U" to "Si" and "T" to "C". Please note, however, that the former Tab. 4 does not exist any more in the revised manuscript because the results of the model validation (over the validation period January 1 to September 1, 2025) are now presented in the new section 3.2 in the context of the new Fig. 3. Instead, the new Tab. 3 presents the ranges of S-Si-C contents from the soil map, as well as the S-Si-C contents derived from that map for the *uncalibrated* SWAP model. For the sake of comprehensiveness, the former Tab. 4 was moved to the supplementary material.

RC: *[2.24] In section 3.2 you describe the possible impact of vegetation dynamics on the CRNS SM. This is something that needs to be discussed more in detail. There have been studies using a dynamic vegetation parameterization in the CRNS soil moisture estimation. Especially in crop- and grasslands this might improve your estimates.*

AR: This might be a misunderstanding. In ll. 207-210 in section 3.2, we discuss the potential effect of vegetation on the SWAP model performance, not on the CRNS-based soil moisture estimation. In the corresponding lines, we had noted a discrepancy between simulated and observed soil moisture in autumn 2024, and had hypothesised that, in the model, the seasonal LAI development for grassland could be a reason for that discrepancy. Meanwhile, we could rule out this reason by testing the sensitivity of this discrepancy to the LAI in that period (not shown). Please also note that the overall discussion of the CRNS results in section 3.1 has changed, and that a detailed comparison of simulated and observed time series is not carried out any more (because we now also compare the observations to the bias-corrected large-scale products, see also comment [1.1]).

Apart from that, the referee is of course right about the effect of vegetation biomass on the CRNS-based soil moisture estimation. However, Heistermann et al. (2024) have also shown that for grassland and cropland sites, the uncertainty originating from vegetation biomass is rather low in comparison to other sources of uncertainty. This is why we assumed a constant biomass for CRNS-based soil moisture estimation in the general calibration framework. In the revised manuscript, we now briefly discuss the potential effects of vegetation biomass on the uncertainty of CRNS-based soil moisture estimates.

RC: *[2.25] Figure 2: it may help to add the land cover type in brackets behind the station name.*

AR: We have added the land cover types in brackets. Please note that Fig. 2 has further changed subject to the revisions: it does not show the simulated time series of SWC any more, since this is now shown in the new Fig. 4, together with the time series of the bias-corrected large-scale products.

RC: *[2.26] The section on "Increasing temporal coverage" is very minimal. I do not see how this is relevant in context with the introduction of a CRNS network for drought monitoring and what the benefit is of your model. Because what you do not show is, how reliable the SM data is for the other years. Is this better then for example satellite data or other models? Likely you do not have validation data from point-scale*

*sensors, but you could compare it to e.g. ERA5-Land and CGLS Soil Water Index in a triple collocation approach to showcase the quality of your data in comparison to existing data sets. In addition, the sections on vertical integration and reconstruction of water fluxes is too minimalistic. At least make a comparison to ERA5-land. Your sites are far enough apart to allow for this.*

AR: In the revised manuscript, we have introduced a new subsection 3.3 in which we discuss the results of subsections 3.1 (CRNS-based soil moisture estimation) and 3.2 (Evaluation of soil moisture products) with regard to potential implications for the management of water-related risks in Brandenburg. "Increasing the temporal coverage" (or "temporal upscaling", as we term it in the revised version) is an important aspect for (drought) risk management or decision support as these typically require to put the soil moisture level at a specific point in time in context with the statistical properties over longer historical periods (typically several decades). This is exactly the motivation of the former Fig. 3 (Fig. 5 in the revised version), and we think it is worthwhile demonstrating this ability in the context of our study and also in the context of the NHESS special issue.

With the formal split-sample calibration and validation, we have validated the ability of the locally calibrated SWAP model to represent soil moisture dynamics at the monitoring locations, and it is of course the motivation of such an analysis to apply the validated model also to time periods outside the calibration/validation frame (in our examples to the period from 1993 to 2024). We have comprehensively revised the manuscript in order to make this clearer, and we hope that the referee will agree that an exemplification of such model-based temporal upscaling (or extrapolation) is worthwhile in the context of section 3.3 (although it necessarily has to remain superficial).

We also much appreciate the referee's suggestion to compare the results of the SWAP model to other available soil moisture products such as ERA5-Land and the CGLS Soil Water Index. We implemented such an analysis as a formal benchmark experiment in which we use the CRNS-based soil moisture estimates as a reference in order to evaluate the performance of various soil moisture products from models and remote sensing. We also visually compare the soil moisture time series of the various products for 2024 and 2025 for all monitoring locations (new Fig. 4). Since this analysis shows that the SWAP model performs better than the large-scale products, we consider it justified to base the two *examples* given in section 3.3 solely on the results of the SWAP model (example 1: volumetric soil water storage for different integration depths and selected years in comparison to the period from 1993 to 2024; example 2: groundwater recharge for selected years in comparison to the period from 1993 to 2024).

[revised manuscript text omitted]

---

## Author Response (AR3)

**Author Response Letter to Editor and Referees**

**Soil moisture monitoring with cosmogenic neutrons: an asset for the development and assessment of soil moisture products in the state of Brandenburg (Germany)**

M. Heistermann, D. Altdorff, T. Francke, M. Schrön, P. M. Grosse, A. Markert, A. Bauriegel, P. Biró, S. Attinger, F. Beyrich, P. Dietrich, R. Eichstädt, P. Grosse, J. Terschlüsen, A. Walz, S. Zacharias, S. Oswald

*NHESS Discussions,* `doi:10.5194/egusphere-2024-3848`
* * *
**EC/RC:** *Editor/referee comment*,     AR: *Author Response*,     ☐ Manuscript text

Dear Dr Somogyvári, dear referees,

we would like to thank you once more for the review of the manuscript. Based on the editorial decision and the referee comments, we have revised the manuscript once more in order to clarify the uncertainty of the CRNS-based soil moisture estimates. Please find below the corresponding comments and a more detailed response.

We would also like to use the opportunity to mention that we updated some information in the manuscript with regard to the new monitoring locations that were meanwhile instrumented (as announced in the manuscript, see ll. 112 of the previous version). The corresponding changes are the following: (1) the monitoring location "Schönhagen" (in the north-east of Brandenburg) had been replaced by the location "Dubrau" (in the south). We changed the corresponding line in table 1 in which we also specify the location attributes, updated Fig. 1 accordingly, and revised all occurrences in the text that referred to "planned" locations; (2) furthermore, the product name of the four new CRNS systems that were deployed by end of 2025 is not "CRS-2000-B" (as previously stated in the four bottom rows of table 1), but "CRS-2000-DE" (according to the manufacturer).

As these new locations were not part of our analysis, we would like to underline that none of these changes affects the scientific results of the paper. They just allow the readers to obtain the latest information about the configuration of the monitoring network.

We hope that the paper can now be accepted for publication.

Thanks again to the editor and the referees for the patience and scrutiny they have invested in this process.

Kind regards,
Maik Heistermann
(on behalf of the author team)

**1. Reply to comments of the editor**

**EC:** *There were still some concerns from reviewers whether the manuscript fits to the scope of NHESS. After some discussion with NHESS editors, we agreed that your contribution is relevant especially for the scope of the special issue.*

**AR:** Thanks for the balanced decision. For us, it was always important to maintain that the manuscript fits well to the scope of the special issue, not necessarily to the overall scope of NHESS (see also our response to referee #5). We are glad that the editorial team agrees.

**EC:** *There are still some outstanding questions regarding the confidence in the results. We would appreciate if you could provide a statement about this in the final submission (please see reviewer report #1).*

**AR:** We have revised the manuscript accordingly. Please see our response to referee #3 below.

**2. Reply to the comments of referee #3**

**RC:** *[...] The reviewer thanks the authors for their extensive work and their well organized answers for the questions brought up during the review. The reviewer specifically thanks the authors for including more/improved context into their manuscript which significantly increased its scientific value and the understanding of the contribution of this network. There is, however, a final remark on the evaluation of the CRNS data, which is central to the methodology of this manuscript. The authors seem to be surprisingly confident about the results of the soil moisture assessment. The reviewer is thankful for the transparency the authors show with respect to their analysis, it, however, delivers a message of mediocre performance comparing the estimated soil moisture date with the evaluated soil moisture data. Originally, the title of the manuscript referred to a drought monitor and the precision the authors show one can raise questions whether this is an approach which would be suitable for such. The CRNS footprint is large and there may be many unknowns of influence (for example the rail tracks or other infrastructure), given the systematic deviation of soil moisture values it may rather required to address the methodology. The Heistermann 2024 et al. approach seems to offer an elegant way to overcome a local calibration, but the manuscript rather seems to confirm that a local calibration as shown by other authors yields better results. At the given stage the reviewer is not having the relevant details to go more deep into the analysis chain of the authors. In good faith the authors may one last time recheck some basic assumptions like the compared count rate or whether the devices would be supplied with a different soil moisture calibration relation in order to potentially get a better alignment with the sample evaluation. Regardless of that technical detail the reviewer is confident about the improvements and congratulates the authors for their article.*

**AR:** We thank the referee for the positive feedback on the revision, as well as for the critical comments. Based on the referee's comment as well as on the editor's suggestion, we revised the manuscript by emphasizing the uncertainties that come along with the CRNS-based soil moisture estimates, in order to avoid the impression of overconfidence in these estimates.

Having said that, we do not fully agree with the impression of being "surprisingly confident about the results of the soil moisture assessment". As the referee stated himself or herself, we spent, in section 3.1, quite an effort to assess the performance of the CRNS-based soil moisture estimation, and also to put it in context with other validation results. On that basis, we referred to the RMSE of 0.037 $m^3\,m^{-3}$ as a "satisfactory agreement" and extensively discussed the role of systematic errors, clearly stating the need to explore the potential of a new conversion function to address this issue. In that section, we also discussed other relevant

sources of uncertainty, and have now, based on the referee's comment, also mentioned (after l. 320 of the previous manuscript version) the potential uncertainty introduced by the proximity to railway tracks (only affects the locations Golm and Paulinenaue).

In order to further address the referee's comment and the editor's suggestion, we have added a statement at the end of section 3.2 (before l. 404 of the previous manuscript version) that the usefulness of the CRNS-based soil moisture estimates as a soil moisture benchmark is of course limited by their own uncertainty (as discussed in section 3.1). We have also put more emphasis on this issue in the conclusions section (after l. 561 of the previous manuscript version).

The referee's question whether the precision of the CRNS-based soil moisture estimates is sufficient for applications in drought monitoring is, in our view, highly dependent on the specific application context and also on the availability of alternative solutions.

Finally, and for the record, we would like to briefly respond to the referee's statement that "[...] the manuscript rather seems to confirm that a local calibration as shown by other authors yields better results". We do not quite agree. In comparison to other studies, our performance metrics are not substantially worse. However, we are aware that a comparison with results from other environments is problematic. The only way to actually confirm such a statement in the context of our study would be the availability of additional groundtruth measurements. That way, we could apply a local calibration on, e.g., the first ground truth measurement, use a second for validation, and compare the performance to the general calibration variant. We do not yet have the data for such a comparison, but hope to obtain them in future research projects. Our honest opinion, however, with regard to the major source of bias, is not the general calibration approach as such, but the use of the conversion function, as repeatedly pointed out in the manuscript. In our opinion, the integration of the conversion function from Koehli et al. (2021) into the general calibration framework should be the way to go in the future.

**3. Reply to the comments of referee #5**

**RC:** *The paper presents an interesting new effort to monitor soil moisture in the state of Brandenburg by developing a network of CRNS stations. It addresses the issue of obtaining reliable, precise soil moisture measurements. The authors describe the design, implementation and evaluation of the new CRNS-network and how these measurements can be useful to improve soil moisture products. The manuscript is well written and has no methodological or major technical flaws. I want to praise the authors and reviewers for the excellent work and great effort put into the manuscript. I agree with the other reviewers that the Brief Communication was not a proper format and that the expanded version better aligns with the authors' intent to publish their technical work. The added analysis and comparison of different datasets also broaden the scope of the papers. Although the paper does not introduce fundamentally new ideas, the authors give meaningful contributions through the documentation of developing a CRNS-network in the state of Brandenburg and how the measurements can be used to improve indirectly-assessed soil moisture. I do believe it is of interest to the scientific community, being particularly useful for other groups interested in developing similar monitoring systems Notwithstanding, it is important to address the limitation of the manuscript's scope. Although the authors consider the context of water-related risks in Brandenburg, the manuscript remains focused on soil-moisture monitoring and technical evaluation of the product, adding limited value to natural hazard discussions. The authors do not directly address natural hazards such as droughts and floods, nor impacts, risks, and policy contexts in the region. Although the paper's focus is on soil moisture monitoring in the State of Brandenburg (Germany), its results and general scope do not tackle*

*natural hazards such as drought and flood risks. It is my opinion that the manuscript should be submitted to a different journal. Recommendation: The manuscript is well-executed and well-written, and deals with the issue of regional monitoring of soil moisture. Readers working on soil moisture measurement, hydrological modelling, and water resources planning/management will have great interest in the text. Given the scope and emphasis of the manuscript, I recommend rejection and advise the authors to consider submitting to a more specialised journal, rather than one focused on natural hazards.*

AR: We appreciate both the positive feedback and the critical appraisal of whether the manuscript fits the journal's scope. With regard to the latter, we fully understand the referee's reservations. However, we are convinced that the specific scope of the special issue would not only include contributions that explicitly address natural hazards research with regard to the federal state of Brandenburg, but which could contribute to any such endeavour. Throughout the manuscript, specifically in section 4, we attempted to outlines perspectives for such contributions.